# Explaining subscription intention for video streaming platforms in China: Integrating the UTAUT2 model, perceived value theory, and S-O-R theory

Tong Wu[1,2,3], Nan Jiang [3], Saeed Pahlevan Sharif[4], Mobai Chen [2]*

**1** Faculty of Media Management, Qingdao Film Academy, Qingdao, Shandong Province, China,
**2** Department of Film and Animation, College of Communication, Qingdao University of Science and Technology, Qingdao, Shandong Province, China, **3** Faculty of Business and Law, Taylor's University, Subang Jaya, Selangor Darul Ehsan, Malaysia, **4** Business School, Sunway University, Subang Jaya, Selangor Darul Ehsan, Malaysia

* 18562637820@163.com & 02604@qust.edu.cn

## Abstract

### Purpose

This study examines the factors influencing the intention to subscribe to video streaming platforms in China through integrating the extended unified theory of acceptance and use of technology (UTAUT2), perceived value theory, and stimulus–organism–response (S-O-R) theory.

### Design/methodology/approach

A quantitative study was conducted to test the hypothesized relationships among the constructs using partial least squares structural equation modeling (PLS-SEM). The sample included 506 viewers of video streaming platforms.

### Findings

The results showed that performance expectancy, effort expectancy, social influence, and hedonic motivation had a positive relationship with subscription intention in video streaming platforms. Additionally, perceived value mediated these relationships. Also, habit was positively related to subscription intention. Finally, the attractiveness of alternatives moderated the relationship between perceived value and subscription intention of video streaming platforms in China.

### Originality/value

This study introduces a moderated mediation mechanism to explain the intention to subscribe to video streaming platforms in China, utilizing the UTAUT2 model, perceived value theory, and S-O-R theory.

which permits unrestricted use, distribution, and reproduction in any medium, provided the original author and source are credited.

**Data availability statement:** All data are available from Harvard Dataverse https://doi.org/10.7910/DVN/OHWKPD

**Funding:** The author(s) received no specific funding for this work.

**Competing interests:** The authors have declared that no competing interests exist.

## Introduction

The global video streaming market has experienced unprecedented growth over the past decade, driven by the rapid proliferation of the Internet, the popularity of mobile devices, and the growing demand for online video content. Subscription video on demand (SVOD) platforms use technologies such as artificial intelligence to align with customers' interests and preferences [1]. These advancements influence consumer subscription behaviors. In particular, the introduction of new business models such as SVOD has significantly impacted consumer behavior towards subscription packages [2–4].

Understanding consumer subscription behaviors is crucial in the context of the evolving video streaming market. This study focuses on subscription intention, which pertains to the degree of a user's inclination to utilize and sustain engagement with video streaming platforms. Research on these concepts links consumers' willingness to pay and their shift from traditional bundles to subscription packages [5,6]. Despite global growth, recent evidence suggests that video streaming platforms in China are facing issues such as high turnover, unstable subscription rates, and difficulty in attracting new customers. Additionally, the overwhelming and confusing information available affects viewers' decisions to subscribe to these platforms [7]. However, the reasons behind these issues remain understudied, highlighting the need to understand the mechanisms informing subscription behavior in the market in China.

Previous research has explored the acceptance of technology, highlighting the roles of factors such as performance expectancy, effort expectancy, social influence, habit, and hedonic motivation in various contexts [8–10]. SVOD is related to the performance expectancy, effort expectancy, social influence, hedonic motivation, and habit that make consumers increase their consumption of these platforms as access is unlimited. According to literature, performance expectancy, effort expectancy, social influence, hedonic motivation, and habit are based on the Extended Unified Theory of Technology Acceptance and Use (UTAUT2) [11]. Furthermore, within different technology acceptance models, this research chooses UTAUT2 as a theoretical framework. According to Venkatesh et al. (2012), the UTAUT theory was proposed to expand the TAM and TAM2 models. Several scholars have presented critiques of the theory, citing mandatory and organizational settings as potential limitations [12]. Venkatesh et al. (2008) revised the theory, resulting in UTAUT2, in response to objections raised by various scholars. And Venkatesh et al. (2008) researched the synthesis of eight commonly used technologies in consumer behavior theory to develop this theory. The fields of psychology, sociology, and human behavior research have made significant advancements in the UTAUT2 theory, considered a cutting-edge theory with minimal errors and imperfections. Thus, UTAUT2 is the most suitable theory for this research and this study chooses UTAUT2 as a theoretical framework [13]. However, there is limited evidence regarding how and under which conditions these factors shape users' subscription intentions to SVOD platforms. Therefore, this study introduces perceived value as a mediator and attractiveness of alternative as a moderator, aiming to enhance our understanding of the mechanisms behind these relationships.

Subscribers' intentions towards video streaming platforms are intricately associated with their perceived value, encompassing convenience, emotional, and social values [3,14,15]. According to the perceived value theory, consumers assess the overall benefits and costs of a service before making a decision [15–17]. Consumers make choices based on their individual preferences, which reflect their underlying motivations [18]. Additionally, the stimulus-organism-response (S-O-R) theory suggests that external stimuli (such as platform features) influence consumers' internal states (i.e., perceived value), which in turn affect their behavioral responses (i.e., subscription intentions) [8,19]. New users are attracted to these platforms by the appeal of their content and the value they perceive during trial periods. As a result, factors such as performance expectancy, effort expectancy, social influence and hedonic motivation are expected to contribute to viewers' perceived value of video streaming platforms, thereby influencing their subscription intentions. Overall, it is important to look into different aspects of how value is perceived. First, the Theory of Consumer Value (TCV) provides a clear delineation of consumption value that helps determine consumers' decision-making intentions [20]. Understanding this delineation is important for video streaming platform providers as they can benefit from understanding why individuals use video streaming platforms and why they choose to watch certain content. In addition, studying the value of consumption can also provide users' service providers with more interactive and engaging content to attract new users and retain existing users [17]. Many studies within the literature use perceived value theory for behavioral intention, especially purchase intention. But most research only looks at how different aspects of how consumers' perceived value affects their purchase decision [21]. Although only the perceived value theory bridges the gap between in-depth research on multi-dimensional perceived value and the impact of perceived value on consumer behavior, there is still a lack of research on the antecedents of perceived value. The effect of performance expectancy, effort expectancy, social influence, hedonic motivation, and habit on subscription intention have not been addressed by using perceived value theory, which is a research gap. In addition, it is a research gap as to whether perceived value has a mediating effect between performance expectancy and subscription intention, effort expectancy and subscription intention, social influence and subscription intention, and hedonic motivation and subscription intention, which needs more research theories to resolve. Furthermore, researchers have recently proposed an integrated S-O-R framework that includes cognitive and emotional systems and all long-term memories from the past [22]. Different scholars expand the S-O-R model by multiple factors in stimulus, organism, and response. Based on the above critical arguments, this research will expand the S-O-R model to apply technology acceptance variables to internal stimuli, i.e., performance expectancy, effort expectancy, social influence, hedonic motivation for viewers. And this research will also expand the organism of S-O-R to perceived value and the response of S-O-R to subscription intention. Additionally, service providers need to understand the factors explaining subscribers' switch to alternative platforms. There is a lot of competition among video streaming platforms. In addition, customer perceptions of the attractiveness of alternatives may be an important factor in attracting customers away from their current service provider [23]. Thus, the moderator of attractiveness of alternative will reflect the real world of the video streaming industry. Therefore, by considering both the perceived value offered by video streaming platforms and the attractiveness of alternatives, this study examines the mechanisms through which performance expectancy, effort expectancy, social influence, hedonic motivation, and habits shape viewers' subscription intentions.

This study contributes to the literature by integrating UTAUT2 model, perceived value theory, and S-O-R theory, providing insights into how and under which conditions performance expectancy, effort expectancy, social influence, hedonic motivation, and habit inform viewers' subscription intentions. Specifically, the study investigates whether the multidimensional perceived value, encompassing convenience, emotional, and social values, explains the processes that inform consumers' subscription intentions. Furthermore, the study explores whether the attractiveness of alternatives mitigates the effects of viewers' perceived value on subscription intentions. In UTAUT2 model highlights the factors of performance expectancy, effort expectancy, social influence, hedonic motivation, and habit in shaping users' subscription intentions. In addition, the multi-dimensional aspects of perceived value are used in this research: emotional value, quality value, monetary value, and convenience value by perceivd value theory. Furthermore, based on S-O-R model, this study explores how

platform features and factors such as performance expectancy, effort expectancy, social influence and hedonic motivation (i.e., Stimulus) influence users' perceived value (i.e., Organism), subsequently impacting their subscription intentions (i.e., Response) on video streaming platforms. Thus, this research integrates UTAUT2, perceived value theory, and S-O-R model to explain the mechanisms informing subscription intention.

The current study focuses on the context of video streaming platforms in China. Over the past few years, video streaming platforms in China have grown continuously. However, increased competition and low barriers to entry may cause customers to switch to other platforms [2]. Few studies have focused on consumer behaviour towards video streaming platforms [18,24], and little research has examined consumers' subscription behaviour [25]. Additionally, in recent years, the unstable subscription numbers of viewers have made it difficult to attract new customers to video streaming platforms in China. For example, from 2020 to 2022, the number of users on IQIYI.COM fell from 107 million to 98.3 million, and Tencent experienced a similar reduction in its subscriber base [7]. Few studies have investigated the reasons for the unstable subscription numbers of viewers, particularly in China. Therefore, this study intends to provide insights into consumer subscription behavior and contribute to the development of effective strategies for retaining and attracting subscribers in the competitive video streaming market in China. Furthermore, new technologies, such as SVOD and artificial intelligence, have changed consumer behavior toward subscription packages in China. SVOD is related to the performance expectancy, effort expectancy, social influence, hedonic motivation, and habit that make consumers to increase their consumption of these platforms as access is unlimited. In addition, consumers in China use these platforms based on their preferences and choices that will help explain their goals. New customers enter these platforms on the basis that they feel attracted to the video content and perceive value during the trial period of the video streaming platform [26]. Thus, this article integrates the UTAUT2 model, perceived value theory, and S-O-R theory to further clarify and contribute to reasons for the unstable subscription numbers of viewers in China. Specifically, this study aims to examine the mediation effect of perceived value on the relationships between performance expectancy, effort expectancy, social influence, and hedonic motivation with subscription intention, as well as to investigate the moderating effect of attractiveness of alternative on the relationship between perceived value and subscription intention for video streaming platforms among Chinese adults.

## Theoretical background

### Unified theory of acceptance and use of technology (UTAUT2) model.

The Unified Theory of Acceptance and Use of Technology (UTAUT2) is widely embraced as a framework for understanding and explaining the acceptance and usage behavior of new technologies [11,27]. Developed by Venkatesh et al. (2003), UTAUT2 integrates insights from eight theories and models related to technology adoption, offering a comprehensive examination of the interplay among key constructs such as performance expectancy, effort expectancy, social influence, facilitating conditions, hedonic motivation, price value, habit, behavioral intention, and technology use [28].

As technologies evolve, the UTAUT2 model is increasingly applied across various technologies and digital platforms, demonstrating its adaptability and ongoing relevance [27,29]. This theoretical framework is particularly useful in understanding the dynamics of emerging digital services such as video streaming platforms [30]. By focusing on the specific user interactions and perceptions within these platforms, researchers can understand the unique challenges and enablers associated with digital media consumption. This adaptability enables a deeper analysis of how, when, and why consumers choose to engage with new technologies, providing valuable insights that drive both academic inquiry and practical application in the digital economy [29].

This study incorporates key variables from the UTAUT2 theory, such as performance expectancy, effort expectancy, social influence, hedonic motivation, and habit, based on relevant literature [11,31], to tailor the model to the specific context of video streaming platforms. Among the UTAUT2 antecedents, performance expectancy is recognized as the most critical factor [29], that would directly influence how viewers perceive the benefits of video streaming platforms, shaping their decision to subscribe.

In the context of video streaming platforms, performance expectancy is defined as the degree to which viewers perceive these platforms as enhancing their performance or aiding in the achievement of their goals [28]. This construct highlights the emphasis consumers place on utilitarian value [29]. It is argued that viewers' intention to subscribe to video streaming platforms is influenced by their perceptions of the benefits associated with adopting and utilizing information technology within these platforms, thereby informing their subscription behavior [32].

The results of the past research on the relationship between performance expectancy and subscription intention are inconclusive. While some studies could not provide support for the relationship between performance expectancy and subscription intention, others found performance expectancy to be a significant predictor of subscription intention [2,9,10,32–37]. Nevertheless, a meta-analysis of 60 studies involving over 122,000 cumulative observations showed that performance expectancy is the most utilized path with the most significant values among all drivers of technology adoption within the UTAUT2 framework [29]. Therefore, while performance expectancy remains a fundamental concept within the UTAUT2 framework in this context, its role in driving subscription behavior on video streaming platforms needs further research. This study states the following hypothesis.

**H1** Performance expectancy positively affects the subscription intention of video streaming platforms.

Effort expectancy, within the scope of this study, refers to the users' perception of how easily they can use video streaming platforms [28]. Given users' preference for simplicity and intuitive design interfaces, a reduction in technological complexity typically results in increased user-friendliness and a smoother user experience, thereby fostering greater adoption of the technology [38]. Similar to the role of performance expectancy, the results of past research on the relationship between effort expectancy and subscription intention are mixed. While some studies have supported its influence on subscription intention [2,33,35,39], others have not provided conclusive evidence [9,10,32,37]. Indeed, a meta-analysis of UTAUT2 studies revealed that, despite being the second most frequently examined predictor, effort expectancy was the weakest of all UTAUT2 relationships [29]. This underscores the need for further research into the role of this construct in explaining users' behavior. This study states the following hypothesis.

**H2** Effort expectancy positively affects the subscription intention of video streaming platforms.

Social influence, within the context of video streaming platforms, refers to the extent to which users perceive significant others, such as family and friends, encouraging their use of such platforms [28]. Consumer preference for adopting information systems can be shaped by the perspectives of their social groups and their willingness to participate in discussions about such systems [2].

While a large body of research has demonstrated a positive impact of social influence on consumers' subscription intention towards video-on-demand services [2,9,32–34,36,39], some could not provide support on this relationship [37]. This study, using the UTAUT2 model, states the following hypothesis.

**H3** Social influence positively affects the subscription intention of video streaming platforms.

Hedonic motivation is defined as the enjoyment and pleasure experienced from using video streaming platforms [11,31]. Venkatesh et al. (2012) suggests that hedonic motivation plays a significant role in shaping users' attitudes and behaviors towards technology adoption [11,40]. Specifically, individuals are more likely to embrace and engage with video streaming platforms when they perceive them as enjoyable and entertaining [41].

This aspect of motivation often stems from the diverse content offerings, interactive features, and immersive experiences provided by these platforms [42,43]. Given the key role of hedonic motivation in driving user engagement, it is essential to examine whether this factor influences the decision-making process of subscribers, especially considering the varying results from past studies [44].

 

The results of recent research on the relationship between hedonic motivation and subscription intention are conflicting. While some studies could not provide support for the relationship between hedonic motivation and subscription intention [9,37], others found hedonic motivation to be a significant predictor of subscription intention [2,32–35,45]. This study states the following hypothesis

**H4** Hedonic motivation positively affects the subscription intention of video streaming platforms.

Tamilmani et al. (2019) describe habit as the most important theoretical addition into UTAUT2 to challenge the role of behavioral intention as the sole predictor of technology use [44]. Limayem et al., (2007) conceptualize the habit of using video streaming platforms as more than just a routine; it encompasses the degree to which a user perceives the utilization of such platforms as automatic and deeply integrated into their daily activities [46].

The subscription model of video streaming platforms relies on frequent video content consumption, meeting user needs, and fostering habitual behavior [47]. Although habit is one source of inconsistency in the UTAUT2 model due to its exclusion in many studies [29], several recent studies have identified habit as a significant predictor of subscription intention [2,9,32–34,36,37,39]. Therefore, the fifth hypothesis is stated as follows.

**H5** Habit positively affects the subscription intention of video streaming platforms.

## Mediating role of perceived value

In the context of video streaming platforms, the UTAUT2 model highlights the roles of factors such as performance expectancy, effort expectancy, social influence, hedonic motivation, and habit in shaping users' subscription intentions [11,29]. However, understanding the mechanism through which these factors influence subscription intentions requires examining mediation and moderation in the model [27].

The concept of consumer perceived value in video streaming platforms pertains to the degree to which users assess the benefits and costs of such platforms about one another. The degree of success and acceptance of a technology or service is contingent upon the value consumers perceive or anticipate from streaming service [48]. And regarding to video streaming platforms in China, viewers' perceived value is very important, which refers to the viewers' comprehensive evaluation of the services, content, experience and other aspects provided by the video streaming platform [49]. For example, in Tencent Video, Iqiyi, Youku and other mainstream video streaming platforms in China, they have continued to enhance the viewers' perceived value by providing high-quality content, optimising the user experience, strengthening their branding, formulating a reasonable pricing strategy, and improving the quality of their services [50]. The multi-dimensional aspects of perceived value are used in this research: emotional value, quality value, monetary value, and convenience value. In this study, emotional value is crucial for video streaming services as it provides a fun or enjoyable service experience to the viewers. In addition, monetary value measures or compares the cost of streaming services with traditional media devices. The quality value measures the video content and video quality which have become increasingly important for viewers to subscribe to video streaming platforms. Furthermore, the convenience value of video streaming services make it convenient, fast, and easy to watch videos on multiple devices, anytime and anywhere. Moreover, the social value of services which through the bullet screen, viewers of video streaming platforms can watch movies and talk to each other at any time [14,15,51].

This study introduces perceived value as the mediator between the driving factors and subscription intention on the basis of the Perceived Value Theory [51] and S-O-R (Stimulus-Organism-Response) model [52]. Perceived Value Theory posits that consumer behavior is influenced by their assessment of the benefits and costs associated with a product or service. In the context of video streaming platforms, consumer-perceived value refers to users' evaluations of the benefits and costs of different platforms, serving as a link between the value proposition offered and users' purchasing behavior [51].

Drawing from the S-O-R (Stimulus-Organism-Response) model [52], which suggests that external stimuli trigger cognitive or emotional responses that lead to behavioral changes, this study explores how platform features and factors such as performance expectancy, effort expectancy, social influence and hedonic motivation (i.e., Stimulus) influence users' perceived value (i.e., Organism), subsequently impacting their subscription intentions (i.e., Response) on video streaming platforms [22]. Despite the theoretical foundation provided by the S-O-R model, there is limited research utilizing it to investigate factors influencing subscription intentions in video streaming platforms.

Existing literature presents conflicting findings regarding the mediating effect of perceived value on these relationships. Some studies have found perceived value to mediate the relationship between performance expectancy, effort expectancy, social influence, hedonic motivation, and habit, and subscription intentions on video streaming platforms [53–59]. However, the results are not unified, and there remains a need for further research to comprehensively understand the impact mechanism of subscription intentions on video streaming platforms.

**H6** Perceived value mediates the relationship between performance expectancy and subscription intention of video streaming platforms.

**H7** Perceived value mediates the relationship between effort expectancy and subscription intention of video streaming platforms.

**H8** Perceived value mediates the relationship between social influence and subscription intention of video streaming platforms.

**H9** Perceived value mediates the relationship between hedonic motivation and subscription intention of video streaming platforms.

**Moderating role of attractiveness of alternatives**

Another contribution of this study is the examination of the moderating role of the attractiveness of alternatives on the relationship between perceived value and subscription intention in the context of video streaming platforms. Investigating this factor is crucial for understanding the dynamics of customer retention and churn in the highly competitive video streaming market, where numerous providers are competing for users' attention.

According to Kuo et al. (2013), the attractiveness of alternatives in video streaming platforms refers to customers' perceptions of the potential for superior services offered by competing video streaming platforms [60]. The extent to which customers perceive viable competitive alternatives in the market is the attractiveness of alternatives. Customer perceptions of the attractiveness of alternatives would play an important role in drawing customers away from their current service provider [23]. Video streaming market in China has formed a diversified competitive landscape, including the co-existence of short-form, long-form and live streaming [61]. In addition, the competition among video streaming platforms in China is extremely fierce, and platforms need to continuously innovate and optimize in terms of content quality, user experience, technological innovation and business models to meet the challenges of market competition. Thus, attractiveness of alternatives in video streaming platforms in China plays an important role in viewers' subscription decisions [62].

The impact of the attractiveness of alternatives on users' decision-making processes has been studied [63] and the literature indicates that the attractiveness of alternatives can significantly influence consumers' behavioral intentions [15,64,65]. However, little is known about its moderating effect in the model that explains users' subscription intention of video streaming platforms.

In this study, the moderation effect of attractiveness of alternatives is important in influencing the subscription intention of video streaming platforms. There are several dimensionals of attractiveness of alternative in this study. VSP alternatives

 

(e.g., TikTok, Bilibili, Slogan, et al.) may offer different packages and attractive international channels. In addition, VSP alternatives that provide different service or function such as trail or different packages. Furthermore, VSP alternatives are more attractive for dual language subtitles and ranking [4,23].

Perceived value is a critical determinant of consumers' subscription intentions, reflecting their assessment of the benefits and costs associated with a video streaming platforms [3,14,15,66,67]. However, this study suggests that the presence of attractive alternatives might alter this relationship. When consumers perceive alternative platforms as offering better or more desirable features, content, or overall service, the perceived value of their current platform may diminish, negatively impacting their subscription intentions.

This study states the following hypothesis.

**H10**  Attractiveness of alternatives moderates the relationship between perceived value and subscription intention of video streaming platforms.

Currently, regarding new media, limited research exists about subscription intention for SVOD, particularly for video streaming platforms in China. There is a lack of academic research on viewers' subscription behavior of video streaming platforms in China. However, it is noted that there is a gap in the current literature in terms of understanding the drivers and determinants of viewers' use of SVOD services [2]. Thus, this research examines the mechanism through factors influence users' subscription decision. However, none of three theories (UTAUT2, perceived value theory, S-O-R model) can comprehensively explain the circumstances in this research. Based on the identified gap in the literature, this study concludes that a single theory is not sufficient to address the current research problem. Thus, this study addresses the gap in existing literature regarding viewers' decision-making process when subscribing to video streaming platforms in China by integration of UTAUT2, perceived value theory, and S-O-R theory.

In the context of video streaming platforms, the UTAUT2 model highlights the roles of factors such as performance expectancy, effort expectancy, social influence, hedonic motivation, and habit in shaping users' subscription intentions [11,27]. Futhermore, this study fills a research gap in the literature, especially regarding the mediating role of perceived value in the relationships between performance expectancy, effort expectancy, social influence, hedonic motivation, and subscription intention in video streaming platforms by the basis of the Perceived Value Theory [51] and S-O-R (Stimulus-Organism-Response) model [3,15,52]. First, this study extends the research based on multi-dimensional perceived value in the context of video streaming platforms by Perceived Value Theory. Secondly, this research expands the S-O-R model to apply the technology acceptance variables to stimulus, which is performance expectancy, effort expectancy, social influence and hedonic motivation for viewers and expands the organism of S-O-R to perceived value. Besides, this research expands the response of S-O-R to subscription intention. Moreover, this research examines the moderating effect of the attractiveness of alternatives on the relationship between perceived value and subscription intention [68,69] (Fig 1).

## Methodology

This study employed a cross-sectional, questionnaire-based approach. For this study, a sample comprising 506 viewers of video streaming platforms in China was obtained using a combination of volunteer (self-selection), and convenience sampling methods. First of all, the sample size is large enough to cover the research question. Secondly, in the questionnaire, it has screening questions to filter out the issues of non-compliance. In addition, the age group is above 18, which can be covered. Furthermore, for viewers' demographic, region and employment status, these two sampling technique has been assessed. Thus, in this study combination of volunteer (self-selection) and convenience sampling methods can be applied to this research. Furthermore, previous scholars are also adopt the similar sampling techniques in their studies [8,15].

Volunteer sampling is the self-selection of sample members to participate in a study. Furthermore, self-selection sampling is when the study allows each case (usually an individual) to determine their willingness to participate in the study

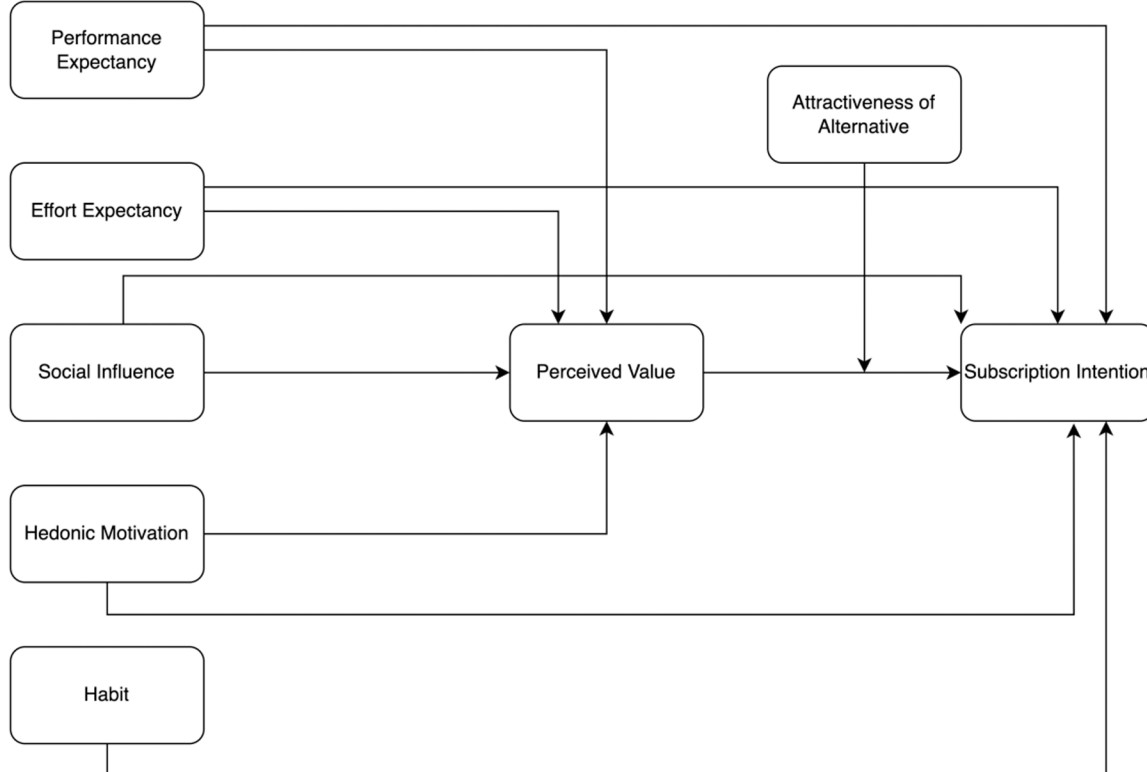

**Fig 1. Research model.**

[70]. Self-selection cases are often selected because they have strong feelings or opinions about the research questions or stated goals. Therefore, the study publicized the need for cases through appropriate media advertising or by inviting volunteers to participate. Thus, it chose this sampling technique because some filtering questions (stated in the appendices of questionnaire) were applied to select the most appropriate samples from the target population in order to meet the researcher's interests regarding the objectives of this study. Additionally, convenience sampling can be used when studying issues such as psychological mechanisms and processes where individual differences are small, the total population is unknown, or random sampling is difficult. This study investigates the influencing factors of subscription intention for VSPs in China, especially about the exploration of the psychological dimension of consumers. Furthermore, for larger survey samples, it is not easy to obtain a complete sample frame. Thus, this study chose convenience sampling to send the questionnaire link as a post through three major social media platforms to facilitate respondents to answer and submit the survey. For sample size, in this research, it uses G*Power (a power probability=0.95, effect size=0.15 (medium), alpha error probability=0.05, number of predictors is 8, the minimum sample size for this study was determined to be 160 respondents [71]. Furthermore, according to Comrey and Lee (2013) conducted a study to assess the sufficiency of sample size for factor analysis, which concludes that an excellent factor analysis program would require a sample size of 1,000 while a sample size of 500 would be considered good [72].

Recruitment of participants occurred through three prominent social media platforms—WeChat, Weibo, and TikTok—as well as three major video streaming platforms in China—Tencent Video, iQIYI, and Youku Video—during the period of October to December 2023. Interested individuals were directed to a questionnaire hosted on the Sojump platform via links distributed across the aforementioned social media platforms. Participation in the study was optional. Eligible

participants were required to be 18 years old or above and to be existing subscribers or potential subscribers of video streaming platforms in China. Furthermore, additional information regarding the ethical, cultural, and scientific considerations specific to inclusivity in global research is included in the Supporting Information (Appendix 1 Checklist). This study is granted permissions by Human Ethics Committee, School of Communication, Qingdao University of Science and Technology.

At the outset of the questionnaire, participants were provided with an explanation of the study's objectives and assured of the confidentiality and anonymity of their responses. Subsequently, respondents were directed to complete the main questionnaire, which consisted of two sections. The first section gathered socio-demographic information such as gender, age, employment status, education level, and monthly income, while the second section comprised items designed to measure the variables delineated in the research model.

### Inclusivity in global research

Additional information regarding the ethical, cultural, and scientific considerations specific to inclusivity in global research is included in the Supporting Information (Appendix 1 Checklist). This study was granted approval by the Human Ethics Committee (HEC), School of Communication, Qingdao University of Science and Technology, China.

### Participants

The respondents comprised 255 males (50.4%) and 251 females (49.6%) who were viewers of video streaming platforms. The majority of participants were full-time employed (78.3%), with 6.9% being part-time employed, 4.9% unemployed, and 3.2% retired, while 6.7% fell into other categories. Regarding education level, most respondents (210, 42.9%) were pursuing a bachelor's degree, 64 (12.6%) were enrolled in vocational school or other programs, and 91 (18%) and 134 (26.5%) were pursuing Master's and Diploma programs, respectively. Concerning monthly allowance/income, respondents reported income levels below RMB3,000 (8.1%), followed by between RMB3,001 and RMB5,000 (14.8%), within the range of RMB5,001 to RMB8,000 (19.6%), and between RMB8,001 and RMB10,000 (29.2%). The remaining respondents reported an income above RMB10,000 (28.3%). Table 1 presents the demographic profile of the participants.

### Measures

The measurement scale employed in this study encompasses several constructs adapted from reputable sources. Subscription intention was measured using eight items, while performance expectancy consisted of nine items, and the construct of effort expectancy comprised nine items, all adapted from Venkatesh et al. Social influence was assessed with eight items, adapted from Venkatesh et al. (2012) [11]. Hedonic motivation was assessed with seven items adapted from Kim et al. (2006) and Venkatesh et al. (2012) [11,31]. Habit was measured with seven items adapted from Limayem, Hirt & Cheung.(2007) and Venkatesh et al. (2012) [11,46]. Perceived value was evaluated using eight items adapted from Zhang et al. (2024), Sweeney and Soutar (2001), and Lee and Overby (2004) [51,73]. Lastly, the attractiveness of alternatives was assessed with nine items adapted from Kuo et al. (2013), Jones et al. (2000), and Kim et al. (2011) [60,74]. Each item of the scale was rated on a seven-point Likert scale ranging from 1 (strongly disagree) to 7 (strongly agree). Additionally, the measurement scale is adopted from previous studies, which is considered valid indicated [11,21,31,46,51,60,73,74] in Appendix 2 Questionnaire. In this study, we did construct validity through 2 marketing experts from video stream industries, 3 university scholars, and 6 potential respondents (randomly selected) to ensure the clarity and comprehensibility of the questionnaire prior to pilot test. Then, this study proceed to pilot study. In addition, digital media consumption and subscription has become a global (e.g., Netflix, HBO Max, IQIYI) tread that is more familiar with most consumers. And this study conducted a final pilot test of the questionnaire draft with 72 users of video streaming platforms in China to assess the construct items analysis, validity analysis, and reliability analysis [75], which indicated all results considered satisfactory.

**Table 1. Demographic Profile of Respondents (N = 506).**

|  | Category | Frequency | Percentage (%) |
|---|---|---|---|
| Gender | Male | 255 | 50.4% |
|  | Female | 251 | 49.6% |
| Age Group | Below 21 | 60 | 11.9% |
|  | 21-28 | 173 | 34.2% |
|  | 29-35 | 129 | 25.5% |
|  | 36-45 | 84 | 16.6% |
|  | 46-59 | 44 | 8.7% |
|  | Above 60 | 16 | 3.2% |
| Employment Status | Full Time Employed | 396 | 78.3% |
|  | Part Time Employed | 35 | 6.9% |
|  | Unemployed | 25 | 4.9% |
|  | Retired | 16 | 3.2% |
|  | Others | 34 | 6.7% |
| Education Level | Senior High School or Below | 64 | 12.6% |
|  | College-Diploma | 134 | 26.5% |
|  | University-Degree | 217 | 42.9% |
|  | Master and above | 91 | 18% |
| Monthly allowance/Income | Below RMB3000 | 41 | 8.1% |
|  | RMB3001–5000 | 75 | 14.8% |
|  | RMB5001–8000 | 99 | 19.6% |
|  | RMB8001–10000 | 148 | 29.2% |
|  | Above RMB10000 | 143 | 28.3% |
| Total |  | 506 | 100% |

## Data Analysis

The initial dataset was corrected by removing missing data and then tested for skewness and kurtosis. After evaluating these estimates, the data were confirmed to be normally distributed, as all values complied with the recommended limits [76]. In addition, in the model fit, the value for SRMR less than 0.08 and NFI above 0.9 are considered a good fit [77]. And in this study, the result of SRMR and NFI are 0.035 and 0.888, which indicated a good model fit (Table 2).

The assessment of the measurement model was conducted using confirmatory factor analysis (CFA) to evaluate composite reliability and convergent validity. Reliability and construct validity were assessed through Cronbach's alpha, factor loadings, composite reliability, and average variance extracted (AVE). The primary measures used in PLS-SEM are Cronbach's alpha (CA) and the composite reliability (CR) value, both considered reliable for values greater than 0.7. To establish convergent validity, the minimum acceptable AVE is 0.5, with values over 0.70 considered satisfactory [78].

Discriminant validity was assessed using two methods: the Fornell-Larcker criterion and the heterotrait-monotrait ratio (HTMT) of correlations. According to the Fornell-Larcker criterion, the AVE of each construct should be greater than its respective maximum shared variance [79]. The HTMT method, with values below 0.9, suggests the presence of discriminant validity [80,81].

The structural model was assessed using partial least squares structural equation modeling (PLS-SEM) with SmartPLS 4.0 software. PLS-SEM is a variance-based method used to analyze complex models containing many latent constructs, indicator variables, and structural paths [82–84]. This method was chosen because it maximizes the variance explained by the model and allows for the estimation of indirect relationships. Given the study's aim to explore theoretical extensions rather than confirmations, PLS-SEM was deemed appropriate [82,83].

**Table 2. Model fit.**

|  | Saturated model | Estimated model |
|---|---|---|
| SRMR | 0.031 | 0.035 |
| d_ULS | 2.068 | 2.694 |
| d_G | 1.038 | 1.048 |
| Chi-square | 2669.424 | 2682.135 |
| NFI | 0.888 | 0.888 |

To address common method variance, the study followed Podsakoff et al. (2003)'s guidelines in designing and administering the questionnaire. Harman's one-way test and marker variable techniques were used to statistically assess common method variance [85]. The results indicated that the first factor accounted for only 29.079% of the cumulative variance, and all *p*-values for the marker variable and constructs were greater than 0.05, indicating insignificant relationships. Thus, common method variance was not a problem in this study.

The structural model was further evaluated for potential collinearity issues using the variance inflation factor (VIF), with a recommended threshold value of 3.3. The structural model's quality was assessed using the Coefficient of Determination ($R^2$), effect size ($f^2$), and predictive relevance ($Q^2$). $R^2$ values were interpreted as 0.26 (substantial), 0.13 (moderate), and 0.02 (weak) [86]. Effect size ($f^2$) values between 0.020–0.150, 0.151–0.350, and greater than 0.350 indicated small, medium, and large effects, respectively [87]. $Q^2$ values above zero indicated predictive relevance, with values between 0.020–0.150, 0.151–0.350, and greater than 0.350 indicating weak, moderate, and strong predictive relevance, respectively [77].

Hypotheses were tested by first evaluating the total effects without the mediator. Subsequently, the mediator was added to the model, and the direct and indirect effects of the mediation model were assessed. The bootstrapping technique with 5000 replications was used to estimate indirect relationships, as it is more accurate and powerful than other methods, such as the Baron and Kenny (1986) and Sobel (1982) approaches [88].

### Ethical consideration

The study protocol and the consent form were approved by the ethical committee of the Human Ethics Committee, author's institution, HEC 2023/298. This study complies with ethical standards by ensuring that all participants were fully informed about the nature and purpose of the research, evidenced by the written informed consent provided on the online distributed questionnaire. An overview of the objectives of the study was provided to the participants, and they were assured that all questionnaires returned were anonymous, and participation in the study was voluntary. Furthermore, in this study the inform consent procedures are robust and participant anonymity is maintained throughout the research process. Finally, this study conducted from the period of 29th October to 30th Decemeber, 2023.

### Results

Table 3 summarizes the results of the measurement model assessment. The results supported construct reliability as well as convergent and discriminant validity of all constructs. Factor loadings ranged from 0.791 to 0.841, thus exceeding the acceptable value of greater than 0.70. Cronbach's alpha (ranged from 0.912 to 0.939) indicated good internal consistency of the items of the constructs. Composite reliability of the constructs (ranged from 0.930 to 0.949) exceeding the acceptable value of 0.70 which supported their construct validity. Convergent validity was assessed using average variance extracted (AVE) of the constructs (ranged from 0.646 to 0.673), which exceeding the acceptable value of 0.50. As the AVE from each construct exceeded the maximum shared variance of the constructs (ranged from -0.531 to 0.517) fulfilling the requirements of Fornell-Larcker standard in discriminant validity (Table 4). Table 5 shows the HTMT matrix demonstrating

**Table 3. Measurement Model Assessment.**

| Construct | Items | Loading | Cronbach's alpha | Composite reliability (rho_a) | Composite reliability (rho_c) | AVE |
|---|---|---|---|---|---|---|
| Attractiveness of alternative | AOA1 | 0.81 | 0.939 | 0.940 | 0.949 | 0.673 |
| | AOA2 | 0.827 | | | | |
| | AOA3 | 0.821 | | | | |
| | AOA4 | 0.833 | | | | |
| | AOA5 | 0.841 | | | | |
| | AOA6 | 0.82 | | | | |
| | AOA7 | 0.821 | | | | |
| | AOA8 | 0.811 | | | | |
| | AOA9 | 0.801 | | | | |
| Effort expectancy | EE1 | 0.804 | 0.936 | 0.938 | 0.946 | 0.663 |
| | EE2 | 0.818 | | | | |
| | EE3 | 0.818 | | | | |
| | EE4 | 0.81 | | | | |
| | EE5 | 0.81 | | | | |
| | EE6 | 0.795 | | | | |
| | EE7 | 0.833 | | | | |
| | EE8 | 0.835 | | | | |
| | EE9 | 0.804 | | | | |
| Habit | H1 | 0.814 | 0.912 | 0.912 | 0.930 | 0.654 |
| | H2 | 0.814 | | | | |
| | H3 | 0.803 | | | | |
| | H4 | 0.819 | | | | |
| | H5 | 0.799 | | | | |
| | H6 | 0.803 | | | | |
| | H7 | 0.81 | | | | |
| Hedonic motivation | HM1 | 0.804 | 0.917 | 0.918 | 0.934 | 0.668 |
| | HM2 | 0.84 | | | | |
| | HM3 | 0.834 | | | | |
| | HM4 | 0.806 | | | | |
| | HM5 | 0.808 | | | | |
| | HM6 | 0.797 | | | | |
| | HM7 | 0.832 | | | | |
| Performance expectancy | PE1 | 0.795 | 0.935 | 0.936 | 0.945 | 0.657 |
| | PE2 | 0.823 | | | | |
| | PE3 | 0.807 | | | | |
| | PE4 | 0.809 | | | | |
| | PE5 | 0.807 | | | | |
| | PE6 | 0.812 | | | | |
| | PE7 | 0.805 | | | | |
| | PE8 | 0.826 | | | | |
| | PE9 | 0.808 | | | | |

*(Continued)*

**Table 3.** (Continued)

| Construct | Items | Loading | Cronbach's alpha | Composite reliability (rho_a) | Composite reliability (rho_c) | AVE |
|---|---|---|---|---|---|---|
| Perceived value | PV1 | 0.807 | 0.922 | 0.923 | 0.936 | 0.646 |
| | PV2 | 0.810 | | | | |
| | PV3 | 0.803 | | | | |
| | PV4 | 0.791 | | | | |
| | PV5 | 0.814 | | | | |
| | PV6 | 0.803 | | | | |
| | PV7 | 0.794 | | | | |
| | PV8 | 0.808 | | | | |
| Subscription intention | S1 | 0.801 | 0.929 | 0.93 | 0.942 | 0.669 |
| | S2 | 0.809 | | | | |
| | S3 | 0.836 | | | | |
| | S4 | 0.806 | | | | |
| | S5 | 0.813 | | | | |
| | S6 | 0.837 | | | | |
| | S7 | 0.819 | | | | |
| | S8 | 0.822 | | | | |
| Social influence | SI1 | 0.795 | 0.926 | 0.927 | 0.939 | 0.659 |
| | SI2 | 0.83 | | | | |
| | SI3 | 0.811 | | | | |
| | SI4 | 0.813 | | | | |
| | SI5 | 0.808 | | | | |
| | SI6 | 0.815 | | | | |
| | SI7 | 0.803 | | | | |
| | SI8 | 0.819 | | | | |

the discriminant validity of the constructs as all the HTMT criteria values were less than 0.9. Table 6 shows that the values of the research model's lateral collinearity assessment are below the threshold of 3.3. Hence, collinearity is not a problem in this study.

The results of the structural model assessment are reported in Table 7. Testing the total effects model indicated that the five antecedents of the model including performance expectancy (PE), effort expectancy (EE), social influence (SI), hedonic motivation (HM) and habit (H) were related to subscription intention (S) providing support for H1 ($\beta = 0.132$,

**Table 4. Fornell-Larcker criterion.**

| | Attractiveness of alternative | Effort expectancy | Habit | Hedonic motivation | Perceived value | Performance expectancy | Social influence | Subscription intention |
|---|---|---|---|---|---|---|---|---|
| Attractiveness of alternative | 0.821 | | | | | | | |
| Effort expectancy | -0.310 | 0.814 | | | | | | |
| Habit | -0.318 | 0.370 | 0.809 | | | | | |
| Hedonic motivation | -0.350 | 0.378 | 0.398 | 0.817 | | | | |
| Perceived value | -0.285 | 0.381 | 0.361 | 0.313 | 0.804 | | | |
| Performance expectancy | -0.270 | 0.225 | 0.282 | 0.306 | 0.240 | 0.810 | | |
| Social influence | -0.307 | 0.410 | 0.337 | 0.330 | 0.336 | 0.234 | 0.812 | |
| Subscription intention | -0.531 | 0.460 | 0.468 | 0.471 | 0.517 | 0.375 | 0.431 | 0.818 |

**Table 5. HTMT Criterion.**

| | AOA | EE | H | HM | PV | PE | SI | S |
|---|---|---|---|---|---|---|---|---|
| Attractiveness of alternative | | | | | | | | |
| Effort expectancy | 0.329 | | | | | | | |
| Habit | 0.343 | 0.399 | | | | | | |
| Hedonic motivation | 0.375 | 0.407 | 0.436 | | | | | |
| Perceived value | 0.303 | 0.407 | 0.393 | 0.338 | | | | |
| Performance expectancy | 0.288 | 0.239 | 0.304 | 0.330 | 0.256 | | | |
| Social influence | 0.329 | 0.440 | 0.366 | 0.358 | 0.361 | 0.252 | | |
| Subscription intention | 0.565 | 0.490 | 0.508 | 0.509 | 0.557 | 0.400 | 0.463 | |

$t = 3.878$, $p < 0.001$), H2 ($\beta = 0.168$, $t = 4.493$, $p < 0.001$), H3 ($\beta = 0.135$, $t = 3.651$, $p < 0.001$), H4 ($\beta = 0.167$, $t = 4.736$, $p < 0.001$) and H5 ($\beta = 0.125$, $t = 3.471$, $p < 0.001$), respectively. The results of testing the indirect effects using bootstrapping approach with 5000 replications indicated that perceived value (PV) mediated the relationship between performance expectancy and subscription intention (H6, $\beta = 0.026$, $t = 2.355$, $p < 0.01$), effort expectancy and subscription intention (H7, $\beta = 0.058$, $t = 4.380$, $p < 0.001$), social influence and subscription intention (H8, $\beta = 0.041$, $t = 3.388$, $p < 0.001$), and hedonic motivation and subscription intention (H9, $\beta = 0.033$, $t = 2.773$, $p < 0.01$), providing support for H6, H7, H8, and H9, respectively. First, mediation analysis was performed to assess the mediating role of PV in the relationship between PE and S, EE and S, SI and S, HM and S. The results (see Table 6) revealed a significant indirect effect of PE on S through PV (H6: $\beta = 0.026$, $t = 2.355$, $p < 0.01$), EE on S through PV (H7: $\beta = 0.058$, $t = 4.380$, $p < 0.001$), SI on S through PV (H8, $\beta = 0.041$, $t = 3.388$, $p < 0.001$), HM on S through PV (H9, $\beta = 0.033$, $t = 2.773$, $p < 0.01$). The total effect of PE on S was significant ($\beta = 0.132$, $t = 3.878$, $p < 0.001$), with inclusion of the mediator the effect of PE on S was still significant ($\beta = 0.106$, $t = 3.277$, $p < 0.01$). The total effect of EE on S was significant ($\beta = 0.168$, $t = 4.493$, $p < 0.001$), with inclusion of the mediator the effect of EE on S was still significant ($\beta = 0.110$, $t = 3.001$, $p < 0.01$). The total effect of SI on S was significant ($\beta = 0.135$, $t = 3.651$, $p < 0.001$), with inclusion of the mediator the effect of SI on S was still significant ($\beta = 0.094$, $t = 2.612$, $p < 0.01$). The total effect of HM on S was significant ($\beta = 0.167$, $t = 4.736$, $p < 0.001$), with inclusion of the mediator the effect of HM on S was still significant ($\beta = 0.134$, $t = 3.845$, $p < 0.001$). This shows a complementary partial mediation role of perceived value in the relationship between PE and S, EE and S, SI and S, HM and S. Hense, H6, H7, H8 and H9 was supported. Finally, the relationship between the interaction of perceived value and attractiveness of alternative with subscription intention was significant ($\beta = -0.087$, $t = 2.607$, $p < 0.01$), which supported H10 on the negative moderating effect of attractiveness of alternative on the association between perceived value and subscription intention.

Figure 2 and Figure 3 shows the research model with the results (Fig 2 and 3). The model explained 21.5% of the variance of perceived value and 54.40% of the variance of subscription intention. In assessment of the structure model, this study examined the coefficient of determination ($R^2$) and predictive relevance ($Q^2$), as reported in Table 8. The results indicated moderate explanatory power of the model for perceived value and substantial explanatory power for subscription intention. Table 9 presents the effect size ($f^2$) of the exogenous constructs in the model, as shown, all $f^2$ values were larger than zero.

**Multi-group analysis.**

In the final part of the research, we evaluated the significant differences between female and male that existed in the effects of influencing factors on subscription intention (S). In the multi-group comparison, gender had a significant difference in the moderation impact of AOA on the relationship between perceived value (PV) and subscription (S). In addition, the differences in path coefficient of moderation (AoA*PV) revealed that female group has relatively greater impact in

**Table 6. Collinearity Assessment.**

|  | AOA | EE | H | HM | PE | PV | S | SI | AOA x PV |
|---|---|---|---|---|---|---|---|---|---|
| AOA |  |  |  |  |  |  | 1.276 |  |  |
| EE |  |  |  |  |  | 1.315 | 1.432 |  |  |
| H |  |  |  |  |  |  | 1.397 |  |  |
| HM |  |  |  |  |  | 1.281 | 1.403 |  |  |
| PE |  |  |  |  |  | 1.136 | 1.195 |  |  |
| PV |  |  |  |  |  |  | 1.323 |  |  |
| S |  |  |  |  |  |  |  |  |  |
| SI |  |  |  |  |  | 1.273 | 1.357 |  |  |
| AOA x PV |  |  |  |  |  |  | 1.049 |  |  |

**Table 7. Structural model assessment.**

| Path | Path coefficient | t-value | p-value | Hypothesis |  | Results |
|---|---|---|---|---|---|---|
| **Moderated Mediation Model** |  |  |  |  |  |  |
| *Indirect Effects* |  |  |  |  |  |  |
| Performance expectancy → Perceived value → Subscription intention | 0.026 | 2.355 | 0.009 | H6 | Partial Mediation | Supported |
| *Direct Effects* |  |  |  |  |  |  |
| Performance expectancy → Subscription intention | 0.106 | 3.277 | 0.001 |  |  |  |
| *Total Effect Model* |  |  |  |  |  |  |
| Performance expectancy → Subscription intention | 0.169 | 4.564 | 0.000 | H1 |  | Supported |
| *Indirect Effects* |  |  |  |  |  |  |
| Effort expectancy → Perceived value → Subscription intention | 0.058 | 4.380 | 0.000 | H7 | Partial Mediation | Supported |
| *Direct Effects* |  |  |  |  |  |  |
| Effort expectancy → Perceived value | 0.236 | 5.185 | 0.000 |  |  |  |
| *Total Effect Model* |  |  |  |  |  |  |
| Effort expectancy → Subscription intention | 0.197 | 5.041 | 0.000 | H2 |  | Supported |
| *Indirect Effects* |  |  |  |  |  |  |
| Social influence → Perceived value → Subscription intention | 0.041 | 3.388 | 0.000 | H8 | Partial Mediation | Supported |
| *Direct Effects* |  |  |  |  |  |  |
| Social influence → Subscription intention | 0.094 | 2.612 | 0.005 |  |  |  |
| *Total Effect Model* |  |  |  |  |  |  |
| Social influence → Subscription intention | 0.173 | 4.496 | 0.000 | H3 |  | Supported |
| *Indirect Effects* |  |  |  |  |  |  |
| Hedonic motivation → Perceived value → Subscription intention | 0.033 | 2.773 | 0.003 | H9 | Partial Mediation | Supported |
| *Direct Effects* |  |  |  |  |  |  |
| Hedonic motivation → Subscription intention | 0.134 | 3.845 | 0.000 |  |  |  |
| *Total Effect Model* |  |  |  |  |  |  |
| Hedonic motivation → Subscription intention | 0.205 | 5.628 | 0.000 | H4 |  | Supported |
| *Total Effect Model* |  |  |  |  |  |  |
| Habit → Subscription intention | 0.208 | 5.338 | 0.000 | H5 |  | Supported |
| *Moderating Effect* |  |  |  |  |  |  |
| Attractiveness of alternative → Subscription intention | -0.275 | 7.743 | 0.000 |  |  |  |
| Perceived value * Attractiveness of alternative → Subscription intention | -0.087 | 2.607 | 0.005 | H10 |  | Supported |

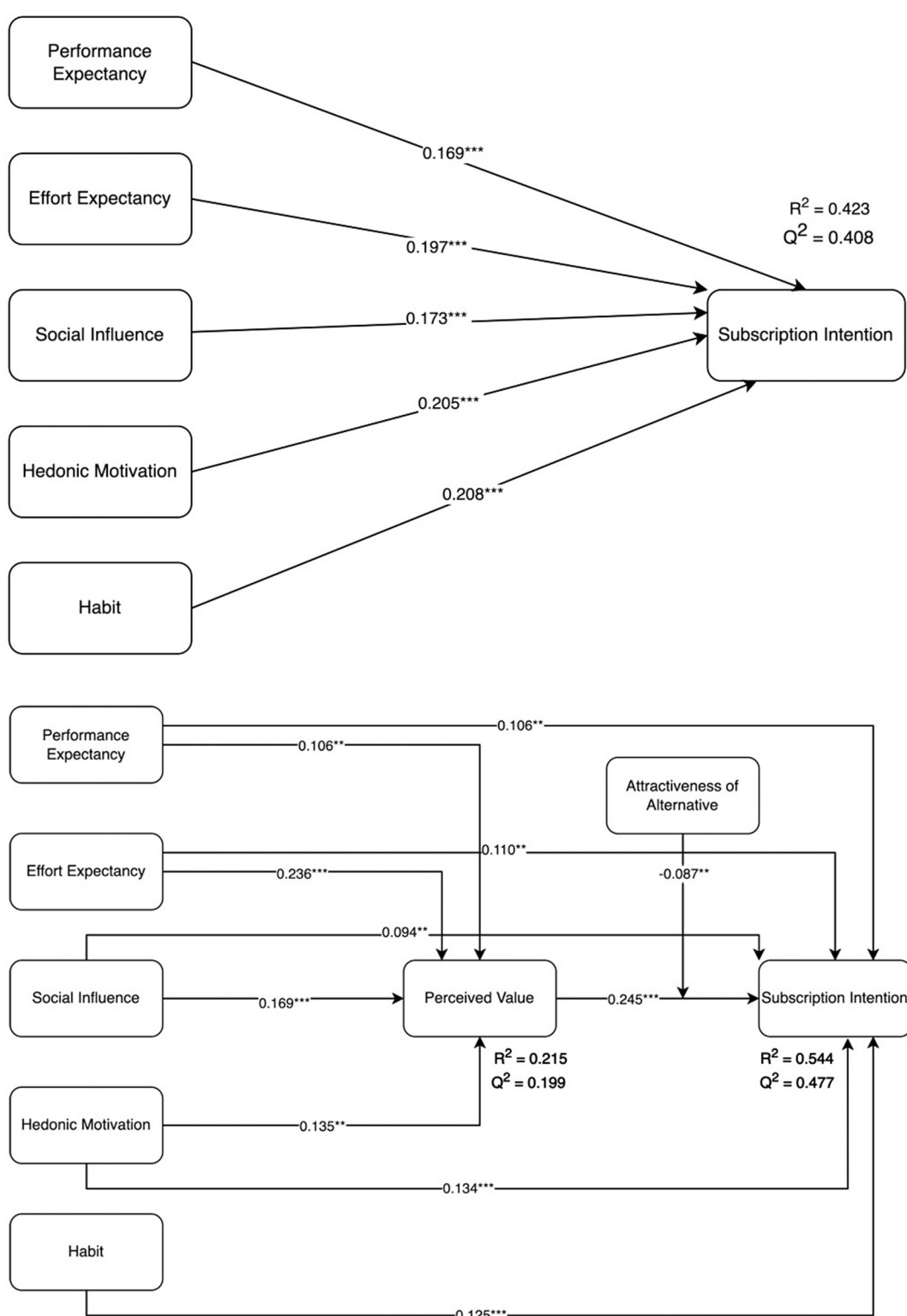

**Fig 2. The structural model with statistically significant paths.**

▲ p < .1, * p < .05, ** p < .01, *** p < .001, two-tailed test

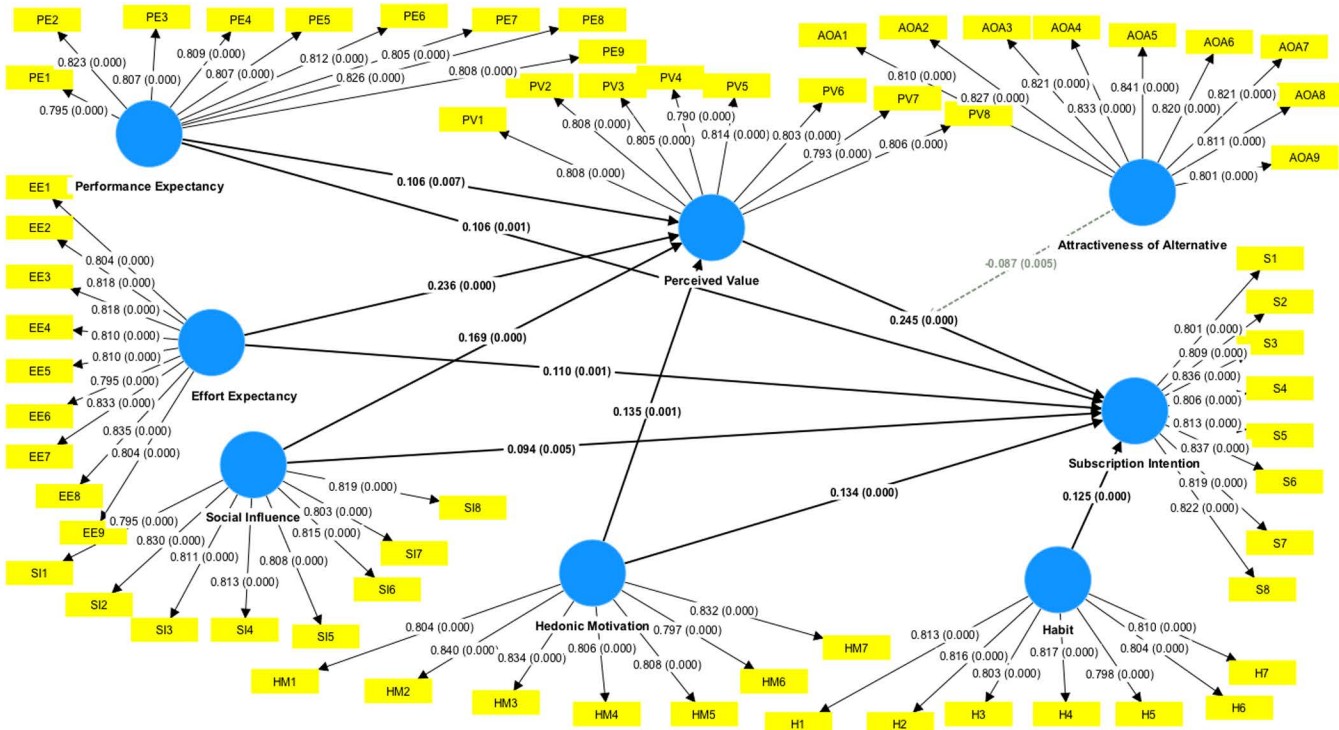

**Fig 3. Research model with path coefficients (outer loadings) and R² values with moderating variable interaction terms.**

**Table 8. R² and Q² values of endogenous variables (With moderating variable indicator).**

| Constructs | *R²* | *Q²* |
|---|---|---|
| Perceived value | 0.215 | 0.199 |
| Subscription intention | 0.544 | 0.477 |

**Table 9. Effect Size (*f* 2) of Exogenous Constructs.**

| Exogenous construct | Relationships | f-square |
|---|---|---|
| Attractiveness of alternative | Attractiveness of Alternative ◊ Social influence | 0.015 |
| Effort expectancy | Effort expectancy ◊ Perceived value | 0.054 |
| | Effort expectancy ◊ Subscription intention | 0.018 |
| Habit | Habit ◊ Subscription intention | 0.025 |
| Hedonic motivation | Hedonic motivation ◊ Perceived value | 0.018 |
| | Hedonic motivation ◊ Subscription intention | 0.028 |
| Perceived value | Perceived value ◊ Subscription intention | 0.099 |
| Performance expectancy | Performance expectancy ◊ Perceived value | 0.013 |
| | Performance expectancy ◊ Subscription intention | 0.021 |
| Social influence | Social influence ◊ Perceived value | 0.029 |
| | Social influence ◊ Subscription intention | 0.014 |

comparison to male group (path diff = 0.135, p = 0.023). All the other differences in the hypothesized relationships were found insignificant. The results of the multi-group analysis are summarized in table 10.

## Discussion

This study aimed to assess the association between performance expectancy, effort expectancy, social influence, hedonic motivation, habit and subscription intention. Additionally, it examined the mediating role of perceived value and the moderating role of the attractiveness of alternatives in this relationship among viewers of video streaming platforms in China.

The findings indicate a positive relationship between performance expectancy and subscription intention (H1). This demonstrates that viewers' perceptions of the usefulness of subscribing to video streaming platforms are closely related to their satisfaction with these platforms in meeting their needs. When users perceive significant benefits from using a video streaming platform to watch videos, they are more likely to exhibit positive subscription intentions. This result is consistent with most previous studies, which have shown that users perceive these platforms as enhancing their performance, thereby increasing their subscription intentions [2,9,10,33,36,37,89]. This consistency underscores the robustness of the performance expectancy construct within the UTAUT2 model, affirming its critical role in shaping user behavior towards technology adoption. However, some studies, such as Yoelianto & Tjhin (2022), did not find a significant relationship between performance expectancy and subscription intention. These discrepancies might be attributed to different study contexts, such as the unique challenges and opportunities in the SVOD market in Indonesia. Factors such as market maturity, cultural differences, and varying levels of technological advancement can all influence performance expectancy and its impact on subscription intentions [90].

Effort expectancy was also found to have a significant positive relationship with subscription intention (H2). This finding demonstrates that the ease of use of video streaming platforms plays a crucial role in influencing users' intentions to subscribe. When the platform interface is user-friendly and easy to navigate, it enhances the overall user experience. Users are more likely to subscribe when they find the platform simple to understand and operate. This result aligns with previous studies that have reported a positive correlation between effort expectancy and subscription intention [2,33,37,89]. These studies emphasize that lower perceived effort in using video streaming platforms increases subscription intentions. However, some studies have not found a significant relationship between effort expectancy and subscription intention [4,36]. These discrepancies may stem from differences in platform design and user experience expectations across various regions and markets.

Table 10. Multi-group analysis.

| Relationships | Difference (Female - Male) | P- value |
|---|---|---|
| EE -> PV | -0.037 | 0.347 |
| EE -> S | -0.045 | 0.273 |
| H -> S | 0.039 | 0.292 |
| HM -> PV | -0.061 | 0.248 |
| HM -> S | 0.000 | 0.499 |
| PE -> PV | -0.049 | 0.292 |
| PE -> S | 0.033 | 0.306 |
| PV -> S | 0.024 | 0.361 |
| SI -> PV | -0.054 | 0.272 |
| SI -> S | 0.096 | 0.097 |
| AOA x PV -> S | 0.135 | 0.023* |

Note: *The Differences are significant in the relationships between female and male (P < 0.05).

The study found a positive relationship between social influence and subscription intention (H3). This indicates that the perceptions and usage patterns of family members, friends, and colleagues directly impact viewers' intentions to subscribe. This finding highlights the importance of social dynamics in shaping consumer behavior towards video streaming services. When individuals see those around them subscribing to and using these platforms, they are more likely to follow suit. This result is consistent with the majority of previous studies, which have reported that the subscription intentions of video streaming platforms are significantly influenced by the social behavior of friends and family members who also subscribe to and watch videos on these platforms [2,8,9,33,34,39,89,91]. However, some studies did not find support for the relationship between social influence and subscription intention [4,36]. These inconsistencies may be due to varying cultural norms and social structures in different regions, which can affect the extent and manner in which social influence impacts subscription decisions.

The study found a positive relationship between hedonic motivation and subscription intention (H4). This indicates that the pleasure derived from using video streaming platforms significantly influences users' intentions to subscribe. The enjoyment and entertainment provided by these platforms are key factors driving subscription intentions. This finding aligns with most previous studies, which have identified hedonic motivation as a key factor driving subscription intentions. The pleasure derived from fulfilling one's entertainment needs through video streaming platforms has been shown to have a significant and positive relationship with subscription intention [2,10,33,37,47,89]. These studies underscore the importance of providing content that users find enjoyable and engaging to enhance their subscription intentions. However, some studies have not found support for the relationship between hedonic motivation and subscription intention [4,36]. These inconsistencies could be attributed to differences in content preferences and entertainment value perceptions across various markets. For instance, the type of content available and cultural attitudes towards leisure and entertainment may vary significantly between countries, influencing the degree to which hedonic motivation impacts subscription intentions [92]. Additionally, the availability of alternative entertainment sources, such as local television or other streaming services, can also affect the relative importance of hedonic motivation in different contexts [93].

Moreover, the results show a positive relationship between habit and subscription intention (H5). This demonstrates that as users develop the habit of watching videos, movies, and programs on video streaming platforms, this behavior becomes an integral part of their lives, significantly increasing their likelihood of subscribing. This finding is consistent with most previous studies that found habit to be an important determinant in capturing users' attention and shaping their behavioral intentions toward digital platforms [2,9,10,15,33,36,37,42,65,89]. However, some authors could not provide support for the relationship between habit and subscription intention [54]. These discrepancies may arise from different user engagement patterns and viewing habits in different contexts. For example, in markets where traditional TV viewing habits are still strong, or where internet connectivity is inconsistent, the development of habitual behavior towards streaming platforms might be less pronounced [94]. Additionally, cultural differences in daily routines and leisure activities can influence how habits form and impact subscription intentions. In regions with higher variability in daily schedules or less emphasis on digital consumption, the role of habit may be less significant in driving subscription behavior [4].

Furthermore, the findings indicate that perceived value mediates the relationships between performance expectancy, effort expectancy, social influence, hedonic motivation, and subscription intention. This suggests that perceived value plays a crucial role in translating the effects of these antecedents into actual subscription intentions for video streaming platforms. Viewers who perceive video streaming platforms as enhancing their performance (H6) consequently attribute higher value to the platform, leading to a greater likelihood of subscribing. Similarly, when viewers find video streaming platforms easy to use (H7), they perceive higher value in the platform, enhancing their subscription intentions. Additionally, social influence (H8) impacts perceived value, as viewers who see their family, friends, or other influencers using video streaming platforms tend to attribute higher value to the platform, thereby increasing their subscription intentions. Finally, users who derive hedonic benefits from using video streaming platforms (H9) are more likely to attribute higher value to the platform, thus enhancing their subscription intention.

These findings are consistent with previous studies [21,41,55–58], underscoring the robustness of the perceived value construct within the UTAUT2 model. The consistent mediation effect of perceived value across these relationships highlights its critical role in converting various antecedents into actionable subscription intentions. By acting as a mediator, perceived value effectively translates the effects of performance expectancy, effort expectancy, social influence, and hedonic motivation into actual subscription intentions. This suggests that video streaming platforms should focus on delivering high performance, ease of use, social engagement, and hedonic benefits, while also effectively communicating and enhancing the overall value perceived by users.

Finally, the findings of the study indicated that the attractiveness of alternatives negatively moderates the relationship between perceived value and subscription intention (H10). This demonstrates that when competing video streaming platforms (VSPs) are available, if viewers perceive low value in a specific VSP, the attractiveness of these alternatives causes subscription intentions toward the specific VSP to decrease. In other words, the presence of appealing alternatives diminishes the impact of perceived value on subscription intentions. When viewers see better options available, they are less likely to subscribe to a platform they perceive as less valuable. This finding is consistent with most previous studies that have found the attractiveness of alternatives negatively moderates the relationship between perceived value and subscription intention [15,65]. These studies emphasize that the availability of competitive alternatives plays a critical role in shaping consumer behavior, particularly when perceived value is low. The attractiveness of alternatives provides viewers with options that may offer superior features, content, or overall service quality, thereby reducing their intention to subscribe to a less valuable platform [95]. The moderating effect of the attractiveness of alternatives highlights the importance for video streaming platforms to not only enhance their perceived value but also to continuously innovate and differentiate themselves from competitors [69]. In highly competitive markets, platforms must focus on maintaining a high perceived value to counteract the allure of alternatives and retain subscriber interest [96].

## Implications

This study investigated the factors influencing subscription intentions for video streaming platforms in China, addressing gaps in the literature on subscription video-on-demand (SVOD) services. By identifying the drivers and determinants of viewers' use of SVOD services, this research expands the understanding of consumer behavior related to video streaming platforms. It also integrates multiple theories to examine these factors, combining the Extended Unified Theory of Acceptance and Use of Technology (UTAUT2), Stimulus–Organism–Response (S-O-R), and perceived value theory. This comprehensive model contributes to marketing research, particularly in the fields of new media, information systems, psychology, and communication.

The research findings offer several significant implications for video streaming platforms. Firstly, perceived value and hedonic motivation were found to be the primary determinants of subscription intention, followed by habit, effort expectancy, performance expectancy, and social influence. This suggests that platforms should focus on enhancing various features and streaming technologies while elevating the quality and diversity of their content offerings [16]. Such efforts can effectively augment viewers' perceived value and consequently boost their subscription intentions.

These findings also indicate that video streaming platforms should invest in new streaming media technologies, such as proprietary program recommender systems and advanced video streaming transmission technologies, to increase users' performance expectancy [53]. Simplifying VSP functionality to make it easier for consumers to use specific platforms can further enhance effort expectancy. Additionally, creating interface functions that allow viewing across multiple devices and improving recommender system functions can enhance the overall user experience [97,98].

Moreover, platforms can increase viewers' subscription intentions by leveraging social influencers, such as friends, family, online celebrities, and social media groups like WeChat. Improving content quality and variety, including films, sports programs, cartoons, games, variety shows, and live broadcasts, can satisfy users' desires for entertainment and gratification [99,100].

Secondly, the significant implication for video streaming platforms is to become viewer-oriented to better understand viewers' preference and desires. The results of this research suggest that VSPs create consistent quality videos, dramas, or films in order to increase consumer perceived value and, thereby, subscription intention. The findings of this research can assist video streaming platforms in enhancing viewers' perceived value of VSPs [20]. This can be achieved through the implementation of new streaming media technologies, enhanced subscription services allowing ad skipping to improve the video-watching experience, and the incorporation of features such as bullet-screen/ live comments. These improvements can attract a larger audience to subscribe to VSPs. In addition, the results of this research may help video streaming platforms to improve viewers'perceived value, to create interface functions by multiple devices, and to improve the functions of the VSP recommender system [16]. The findings of this research can aid video streaming platforms in enhancing viewers' perceived value of VSPs through influencers such as friends or family members, other online celebrities, and social media. This, in turn, can drive an increase in subscribers to video streaming platforms.

The third significant implication for VSPs is to understand better their viewers' practical concerns and retain or improve their subscription rate. The results of this research may help video streaming platforms understand that improving perceived value for viewers is the key factor to increase subscription intention. In addition, video streaming platforms can improve the quality and variety of content in VSPs to stimulate viewers' hedonic motivation and thereby increase their satisfaction and gratification. Then, more viewers will choose to subscribe [101].

The fourth significant implication for the platforms and industry regards the the negative moderating effect that the attraction of alternatives (AOA) has on subscription intention. If there are too many VSP competitors, then subscription intention will be decreased. Thus, competition should occur between only a few monopoly VSPs rather than many small VSPs [4]. In summary, a smaller number of competing VSPs would increase overall subscription intention. Furthermore, the study highlights the negative moderating effect of the attractiveness of alternatives (AOA) on subscription intention. To counteract this effect, platforms should increase their content variety and quality to enhance competitiveness and perceived value, thereby boosting subscription intentions [97]. In addition, to mitigate the effect of alternate attractiveness of video streaming platform competitors on subscription intention. Video streaming platforms must improve their essential capabilities. Additionally, they utilize user profiles for distinguishing their key user groups [102]. Furthermore, video streaming platforms must give a positive user experience. And, they must have a comprehensive understanding of users' demands and conduct in-depth analyses of its target users' interests, desires, and habits in order to develop content that satisfies their requirements [103].

## Limitations and future research

This research, while providing valuable insights into the factors influencing subscription intentions for video streaming platforms in China, has several limitations that should be acknowledged.

The first limitation pertains to the research scope and geographical focus, which were confined to video streaming platforms in China. The findings may not be generalizable to other countries, as the development and competition of video streaming platforms, as well as cultural preferences, can vary significantly across different regions [104]. Additionally, examining the same hypotheses in different countries may yield different results in terms of different consumer subscription behaviors due to cultural, ethnic, and demographic differences. Cultural nuances strongly influence consumer perceptions and behaviors, suggesting that further research is necessary to test the model in various regions and cultural contexts. Future studies should consider including a more diverse range of participants to represent different cultural backgrounds adequately, ensuring broader applicability of the results [2,8]. Furthermore, in future study, it should be added potential longitudinal studies to assess changes in subscription intention over time.

The second limitation is the exclusive use of quantitative methods to investigate the factors influencing subscription intentions in China. While quantitative data provides a robust statistical analysis, it may not capture the full complexity of consumer behavior. However, in consumer behavior research, quantitative studies may fail to capture the deeper

motivations and specific social contexts of consumer behavior, resulting in findings that lack depth and utility. Thus, future research should consider applying qualitative or mixed methods approaches to gain deeper insights into the underlying motivations and attitudes driving subscription behavior. Qualitative methods, such as interviews and focus groups, can provide rich, detailed data that complement quantitative findings, leading to a more comprehensive understanding of consumer behavior in the dynamic context of digital entertainment [1,105]. Furthermore, in sampling methodology, self-selection, volunteer, and convenience sampling methods are non-probability sampling, which include an element of subjective judgment and cannot infer the overall population quantitatively. Thus, in future research, it can choose to use stratified random sampling to improve the representativeness of the research.

The third limitation is the focus solely on the user's perspective. In addition, relying only on the sample of user groups and the influences of the research framework may lead to biased results, and the influences of video streaming platforms and industry practitioners may also affect the results of the study. This study did not collect data from video streaming platforms or industry practitioners, which could provide valuable insights into the operational and strategic factors influencing subscription intentions. Future research should incorporate perspectives from a diverse range of video streaming platforms and industry professionals to create a new research framework model to gain a holistic view of the factors affecting subscription behavior. Understanding the industry's challenges, strategies, and perspectives can offer a more rounded analysis and help in formulating more effective recommendations for enhancing subscription rates [106,107].

By acknowledging these limitations and exploring these avenues for future research, scholars can improve our understanding of consumer behavior in the evolving digital entertainment landscape. This will enable the development of more comprehensive models that account for diverse perspectives and methodologies, ultimately contributing to a more nuanced understanding of the factors driving subscription intentions for video streaming platforms [15].

## Supporting Information

**S1. Appendix 1: Checklist.**
(DOCX)

**S2. Appendix 2: Questionnaire.**
(DOCX)

## Author contributions

**Conceptualization:** Tong Wu.

**Data curation:** Nan Jiang.

**Formal analysis:** Tong Wu.

**Funding acquisition:** Mobai Chen.

**Investigation:** Saeed Pahlevan Sharif.

**Methodology:** Nan Jiang.

**Project administration:** Mobai Chen.

**Resources:** Mobai Chen.

**Software:** Tong Wu, Nan Jiang.

**Supervision:** Nan Jiang, Saeed Pahlevan Sharif, Mobai Chen.

**Validation:** Saeed Pahlevan Sharif.

**Visualization:** Nan Jiang, Saeed Pahlevan Sharif.

**Writing – original draft:** Tong Wu.

**Writing – review & editing:** Saeed Pahlevan Sharif.

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
