## [Decision Letter · Decision Letter 0]

24 Sep 2024

PONE-D-24-28668Explaining subscription intention for video streaming platforms in China: Integrating the UTAUT2 model, perceived value theory, and S-O-R theoryPLOS ONE

Dear Dr. Chen,

Thank you for submitting your manuscript to PLOS ONE. After careful consideration, we feel that it has merit but does not fully meet PLOS ONE’s publication criteria as it currently stands. Therefore, we invite you to submit a revised version of the manuscript that addresses the points raised during the review process.

The research area is very interesting, but the manuscript needs additional details in multiple sections. Theoretical integration, measurement items and methodology, data analysis transparency in results, results’ discussion and paper’s contributions are needed.

We look forward to receiving your revised manuscript.

Kind regards,

Simona Vinerean, PhD

Academic Editor

PLOS ONE

3. Please ensure that you include a title page within your main document. You should list all authors and all affiliations as per our author instructions and clearly indicate the corresponding author.

Additional Editor Comments:

On behalf of the editorial team, I would like to thank you for submitting your manuscript to PLOS One for publication consideration. The paper is quite interesting for a wide audience. As an additional recommendation to facilitate readability, Table 6 could also include the level of acceptance/ rejection of each hypothesis.

Reviewers have provided extended suggestions; please work on them meticulously and submit the revised version.

Reviewers' comments:

Reviewer's Responses to Questions

**Comments to the Author**

1. Is the manuscript technically sound, and do the data support the conclusions?

Reviewer #1: Yes

Reviewer #2: Partly

2. Has the statistical analysis been performed appropriately and rigorously? 

Reviewer #1: Yes

Reviewer #2: N/A

3. Have the authors made all data underlying the findings in their manuscript fully available?

Reviewer #1: Yes

Reviewer #2: Yes

4. Is the manuscript presented in an intelligible fashion and written in standard English?

Reviewer #1: Yes

Reviewer #2: Yes

5. Review Comments to the Author

Reviewer #1: 1. Theoretical Integration and Clarification: The paper proposes an integration of UTAUT2, perceived value theory, and S-O-R theory. However, the linkage between these theories is not sufficiently clarified. How do these theories synergistically enhance our understanding of subscription intention? Provide a more detailed rationale for this theoretical integration.

2. Operationalization of Constructs: The operational definitions of some constructs, such as perceived value and attractiveness of alternatives, could benefit from further elaboration. How do these constructs operationalize within the context of Chinese video streaming platforms? Clarify and justify the dimensions chosen.

3. Sampling Methodology: The paper uses a combination of self-selection, volunteer, and convenience sampling methods, which could introduce bias. Discuss the potential implications of this sampling strategy on the generalizability of the findings and consider stratified random sampling for future research to enhance representativeness.

4. Measurement Scales: You have adapted items from different sources for your constructs. Please ensure that these items are culturally appropriate for the Chinese market and that any necessary translations are validated.

5. Data Analysis: While PLS-SEM is used, there is an opportunity to deepen the analysis. Consider performing multi-group analysis to explore potential differences in subscription intention across different demographic groups (e.g., age, income level).

6. Mediation Analysis: The mediation effect of perceived value is indicated, but the paper could benefit from a more nuanced exploration of the indirect effects. Specifically, consider the possibility of partial or full mediation and explore this in greater depth.

7. Moderation Analysis: The moderating effect of attractiveness of alternatives is a key contribution of your paper. Provide a more in-depth discussion of the implications of this finding, particularly how service providers might address the challenge of alternative attractiveness.

8. Results Presentation: The results are presented concisely, but the paper could benefit from additional visual aids such as graphs to illustrate key relationships and interactions more clearly.

9. Implications for Practice: The implications for video streaming platforms are mentioned, but they could be more actionable. Provide specific strategies that platforms can use to enhance perceived value and mitigate the impact of attractive alternatives.

10. Limitations and Future Research: The limitations are acknowledged; however, expand on how these limitations could specifically affect the findings of the study. Furthermore, propose clear directions for future research, including potential longitudinal studies to assess changes in subscription intention over time.

11. Ethical Considerations: It is good that ethical considerations have been addressed, but ensure that informed consent procedures are robust and that participant anonymity is maintained throughout the research process.

12. References and Citations: Ensure that all references are current and relevant, and that citations are consistently formatted according to the appropriate academic style guide.

In summary, while the paper makes a valuable contribution to understanding subscription intentions for video streaming platforms in China, these revisions could significantly strengthen the study's theoretical grounding, methodological rigor, and practical implications.

Reviewer #2: It can be acknowledged that the authors research idea is interesting and their research effort consistent.

In the following, I provide some recommendation for improvement:

1. The theoretical background of the research is presented well - UTAUT2, perceived value theory and SOR theory are described alongside with scientific articles using them. Regarding the perceived value theory, the authors state:

As a result, factors such as performance expectancy, effort expectancy, social influence, hedonic motivation, and habits are expected to contribute to viewers' perceived value of video streaming platforms, thereby influencing their subscription intentions (14-16).

Thus, habits are considered a factor for influencing viewers' perceived value of video streaming platforms.

Within 2.2. Mediating Role of perceived value, the authors state:

Drawing from the S-O-R (Stimulus-Organism-Response) model (Mehrabian & Russell, 1974), which suggests that external stimuli trigger cognitive or emotional responses that lead to behavioral changes, this study explores how platform features and factors such as performance expectancy, effort expectancy, social influence, hedonic motivation, and habit (i.e., Stimulus) influence users' perceived value (i.e., Organism), subsequently impacting their subscription intentions (i.e., Response) on video streaming platforms (Jacoby, 28, 2002). Despite the theoretical foundation provided by the S-O-R model...

Thus, once again, the authors consider habit as an influence factor of perceived value.

...but there is no research hypothesis regarding the mediation role of perceived value for the habit - subscription intention relationship.

Maybe the authors should argument why this hypothesis was not included (although, as they mentioned previously, ... and habit (i.e., Stimulus) influence users' perceived value (i.e., Organism)..)

2. Both measurement and structural equation model should be statistically tested through tests/indicators such as:

ChiSquare, RMSEA, PCLOSE and/or NFI,RFI,IFI,CFI and/or GFO,AGFI.

3. Table 5 - Colliniarity Assessment has missing values. The authirs should explain in more detail what this table is about - I can only deduce (maybe wrongly) that the main diagonal elements are SQRAVE values and the other elements are partial correlation coefficient values.

6. PLOS authors have the option to publish the peer review history of their article (what does this mean? ). If published, this will include your full peer review and any attached files.

**Do you want your identity to be public for this peer review?** For information about this choice, including consent withdrawal, please see our Privacy Policy .

Reviewer #1: No

Reviewer #2: No

---

## [Author Response · Author response to Decision Letter 1]

24 Nov 2024

Rebuttal Letter

Dear Editor and Reviewers,

We would like to express our sincere gratitude for your valuable comments and constructive suggestions. Your feedback is very helpful in guiding us through the manuscript amendments. We have resubmitted the updated manuscript for your kind review. Thank you very much for offering us the opportunity to improve our work. Please refer to the marked contents and sections in the 'Revised Manuscript with Track Changes'. Author action of each comment is presented in the following sections:

Thank you

Kind regards

Author Team

Editor’s comments

1.Please ensure that your manuscript meets PLOS ONE's style requirements, including those for file naming.

Author Action: Thanks for your valuable comments. I rechecked the PLOS ONE style requirements and file name, revised title accordingly. I revised the heading size and format, figure files formats (Resubmitted figure file), reference citations format according to MANUSCRIPT BODY FORMATTING GUIDELINES. And I added author information in the first page of manuscript line 1-13. In addition, I revised the format of file of Title_Author_Affiliations and resubmitted again.

2.Please include a complete copy of PLOS’ questionnaire on inclusivity in global research in your revised manuscript.

Author action: Thanks for your valuable comments. I added this study added details as to who granted permissions and/or consent for the study to take place in the section of inclusivity in global research, part of Methodology, line 8-11, which is indicated as “Additional information regarding the ethical, cultural, and scientific considerations specific to inclusivity in global research is included in the Supporting Information (S4_Appendix 1 Checklist). This study was granted approval by the Human Ethics Committee (HEC), School of Communication, Qingdao University of Science and Technology, China.” The HEC approval letter, informed consent form, survey cover page, and the full questionnaire (in both Mandarin and English) are included in S4_Appendix 1: Questionnaire as supporting information.

3.Please ensure that you include a title page within your main document.

Author Action: Thanks for your valuable comments. I added title page in first page of manuscript, line 1-13.

4.As an additional recommendation to facilitate readability, Table 6 could also include the level of acceptance/ rejection of each hypothesis.

Author Action: Thanks for your valuable comments. I added results which indicated supported or not supported in section of 4.0 Results, line 1, table 6.

Reviewer 1 Comments:

1.Theoretical Integration and Clarification: The paper proposes an integration of UTAUT2, perceived value theory, and S-O-R theory. However, the linkage between these theories is not sufficiently clarified. How do these theories synergistically enhance our understanding of subscription intention? Provide a more detailed rationale for this theoretical integration.

Author Action: Thanks for your valuable commets. To address the comments, we revised the entire manuscript. I added the detailed rationale for theoretical integration in the theoretical background section, behind 2.3 Moderating role of attractiveness of alternatives, pharagraph 6, line 4-13, stated as “Currently, regarding new media, limited research exists about subscription intention for SVOD, particularly for video streaming platforms in China. There is a lack of academic research on viewers’ subscription behavior of video streaming platforms in China. However, it is noted that there is a gap in the current literature in terms of understanding the drivers and determinants of viewers’ use of SVOD services [7]. Thus, this research examines the mechanism through factors influence users’ subscription decision. However, none of three theories (UTAUT2, perceived value theory, S-O-R model) can comprehensively explain the circumstances in this research. Based on the identified gap in the literature, this study concludes that a single theory is not sufficient to address the current research problem. Thus, this study addresses the gap in existing literature regarding viewers’ decision-making process when subscribing to video streaming platforms in China by integration of UTAUT2, perceived value theory, and S-O-R theory.” And added contents in theoretical background section, behind 2.3 Moderating role of attractiveness of alternatives, pharagraph 7, line 14-26, stated as “In the context of video streaming platforms, the UTAUT2 model highlights the roles of factors such as performance expectancy, effort expectancy, social influence, hedonic motivation, and habit in shaping users' subscription intentions [84, 90]. Futhermore, this study fills a research gap in the literature, especially regarding the mediating role of perceived value in the relationships between performance expectancy, effort expectancy, social influence, hedonic motivation, and subscription intention in video streaming platforms by the basis of the Perceived Value Theory [79] and S-O-R (Stimulus-Organism-Response) model [19, 54, 91]. First, this study extends the research based on multi-dimensional perceived value in the context of video streaming platforms by Perceived Value Theory. Secondly, this research expands the S-O-R model to apply the technology acceptance variables to stimulus, which is performance expectancy, effort expectancy, social influence and hedonic motivation for viewers and expands the organism of S-O-R to perceived value. Besides, this research expands the response of S-O-R to subscription intention. Moreover, this research examines the moderating effect of the attractiveness of alternatives on the relationship between perceived value and subscription intention [36, 57]. Figure 1 is the research framework.”.

2.Operationalization of Constructs: The operational definitions of some constructs, such as perceived value and attractiveness of alternatives, could benefit from further elaboration. How do these constructs operationalize within the context of Chinese video streaming platforms? Clarify and justify the dimensions chosen.

Author Action: Thanks for your valuable commets. To address the comments, we revised the entire manuscript. I added the contents of definition of perceived value and the constructs operationalization by multiple dimentions, which is added in theoretical background section, 2.2 Mediating role of perceived value, pharagraph 2, line 13-24, stated as “The concept of consumer perceived value in video streaming platforms pertains to the degree to which users assess the benefits and costs of such platforms about one another. The degree of success and acceptance of a technology or service is contingent upon the value consumers perceive or anticipate from streaming service [107]. The multi-dimensional aspects of perceived value are used in this research: emotional value, quality value, monetary value, and convenience value. In this study, emotional value is crucial for video streaming services as it provides a fun or enjoyable service experience to the viewers. In addition, monetary value measures or compares the cost of streaming services with traditional media devices. The quality value measures the video content and video quality which have become increasingly important for viewers to subscribe to video streaming platforms. Furthermore, the convenience value of video streaming services make it convenient, fast, and easy to watch videos on multiple devices, anytime and anywhere. Moreover, the social value of services which through the bullet screen, viewers of video streaming platforms can watch movies and talk to each other at any time [19, 59, 79].”Furthermore, I added the contents of definition of attractiveness of alternatives and the constructs operationalization by multiple dimentions, which is added in theoretical background section, 2.3 Moderation role of attractiveness of alternatives, pharagraph 2, line 20-21, stated as “The extent to which customers perceive viable competitive alternatives in the market is the attractiveness of alternatives.” And theoretical background section, 2.3 Moderation role of attractiveness of alternatives, pharagraph 4, line 27-32, stated as “In this study, the moderation effect of attractiveness of alternatives is important in influencing the subscription intention of video streaming platforms. There are several dimensionals of attractiveness of alternative in this study. VSP alternatives (e.g TikTok, Bilibili, Slogan, et al.) may offer different packages and attractive international channels. In addition, VSP alternatives that provide different service or function such as trail or different packages. Furthermore, VSP alternatives are more attractive for dual language subtitles and ranking [69, 88].”

3.Sampling Methodology: The paper uses a combination of self-selection, volunteer, and convenience sampling methods, which could introduce bias. Discuss the potential implications of this sampling strategy on the generalizability of the findings and consider stratified random sampling for future research to enhance representativeness.

Author Action: Thanks for your valuable comments. To address the comments, we revised the entire manuscript. I added potential implications of this sampling strategy on the generalizability of the findings and consider stratified random sampling for future research to enhance representativeness in section of Limitations and future research, pharagraph 3, line 28-31, stated as “Furthermore, in sampling methodology, self-selection, volunteer, and convenience sampling methods are non-probability sampling, which include an element of subjective judgment and cannot infer the overall population quantitatively. Thus, in future research, it can choose to use stratified random sampling to improve the representativeness of the research.”

And, volunteer sampling technique was chosen by the researcher of this current study, because some filtering questions in Appendix 2, were applied to select the most appropriate samples from the target population, to meet the researcher’s interests regarding the objectives of this study. In convenience sampling, the advantages are convenient, simple, and easy to implement, saving time and money. In addition, convenience sampling can be used when studying issues such as psychological mechanisms and processes where individual differences are small, the total population is unknown, and random sampling is difficult. Furthermore, this study use marker variable to minimize the potential generalization problems in this study.

4.Measurement Scales: You have adapted items from different sources for your constructs. Please ensure that these items are culturally appropriate for the Chinese market and that any necessary translations are validated.

Author Action: Thanks for your valuable comments. I have been rechecked all items of measurement scales to be suitable for Chinese market and rechecked the translations, which is validated. And I added the questionnaire within my manuscript in appendix 1.

5.Data Analysis: While PLS-SEM is used, there is an opportunity to deepen the analysis. Consider performing multi-group analysis to explore potential differences in subscription intention across different demographic groups (e.g., age, income level).

Author Action: Thanks for your variable comments. According to your great advices, this study performed multi-group analysis to explore potential differences in subscription intention across income level, gender and age of different demographic groups. However, I found out the outcome of MGA is insignificant.

For age groups of 18-45 and age above 45, the result of MGA is insignificant.

For gender groups of female and male to run MGA, the result indicated the multi-group analysis is insignificant.

For different income level of demographic groups, the result of MGA is insignificant.

6.Mediation Analysis: The mediation effect of perceived value is indicated, but the paper could benefit from a more nuanced exploration of the indirect effects. Specifically, consider the possibility of partial or full mediation and explore this in greater depth.

Author action: Thanks for your variable comments. To address the comments, I deleted the contents in Results section, the part of mediation analysis, phragraph 2, line 27-31, which indicated as “Considering the significant relationship between performance expectancy (β = 0.106, t = 3.277, p < 0.01), effort expectancy (β = 0.110, t = 3.001, p < 0.01), social influence (β = 0.094, t = 2.612, p < 0.01), and hedonic motivation (β = 0.134, t = 3.845, p < 0.001) with subscription intention in the moderated mediation model, perceived value partially mediate these relationships.”. And I further added the contents of detailed explaination of indirect effects and consider the possiblity of partial or full mediation to further investigate it, which in Results section, the part of mediation analysis, phragraph 2, line 31-42, which indicated as “First, mediation analysis was performed to assess the mediating role of PV in the relationship between PE and S, EE and S, SI and S, HM and S. The results (see Table 6) revealed a significant indirect effect of PE on S through PV (H6: β = 0.026, t = 2.355, p < 0.01), EE on S through PV (H7: β = 0.058, t = 4.380, p < 0.001), SI on S through PV (H8, β = 0.041, t = 3.388, p < 0.001), HM on S through PV (H9, β = 0.033, t = 2.773, p < 0.01). The total effect of PE on S was significant (β = 0.132, t = 3.878, p < 0.001), with inclusion of the mediator the effect of PE on S was still significant (β = 0.106, t = 3.277, p < 0.01). The total effect of EE on S was significant (β = 0.168, t = 4.493, p < 0.001), with inclusion of the mediator the effect of EE on S was still significant (β = 0.110, t = 3.001, p < 0.01). The total effect of SI on S was significant (β = 0.135, t = 3.651, p < 0.001), with inclusion of the mediator the effect of SI on S was still significant (β = 0.094, t = 2.612, p < 0.01). The total effect of HM on S was significant (β = 0.167, t = 4.736, p < 0.001), with inclusion of the mediator the effect of HM on S was still significant (β = 0.134, t = 3.845, p < 0.001). This shows a complementary partial mediation role of perceived value in the relationship between PE and S, EE and S, SI and S, HM and S. Hense, H6, H7, H8 and H9 was supported.”

7.Moderation Analysis: The moderating effect of attractiveness of alternatives is a key contribution of your paper. Provide a more in-depth discussion of the implications of this finding, particularly how service providers might address the challenge of alternative attractiveness.

Author action: Thanks for your valuable comments. I added more in depth discussion of the implications of this finding in the part of Implications, pharagraph 5, line 20-25, which is indicated as “In addition, the results of this research may help video streaming platforms to improve viewers’perceived value, to create interface functions by multiple devices, and to improve the functions of the VSP recommender system [74]. The findings of this research can aid video streaming platforms in enhancing viewers' perceived value of VSPs through influencers such as friends or family members, other online celebrities, and social media. This, in turn, can drive an increase in subscribers to video streaming platforms.”

8.Results Presentation: The results are presented concisely, but the paper could benefit from additional visual aids such as graphs to illustrate key relationships and interactions more clearly.

Author action: Thanks for your valuable comments. To address the comments, I added the Figure 3: Research model with path coefficients (outer loadings) and R² values with moderating variable interaction terms in section of Results.

9. Implications for Practice: The implications for video streaming platforms are mentioned, but they could be more actionable. Provide specific strategies that platforms can use to enhance perceived value and mitigate the impact of attractive alternatives.

Author action: Thanks for your valuable comments. I added strategies that platforms can use to enhance perceived value to increase subscription intention in section of implications, pharagraph 5, line 13-

---

## [Decision Letter · Decision Letter 1]

5 Feb 2025

PONE-D-24-28668R1Explaining subscription intention for video streaming platforms in China: Integrating the UTAUT2 model, perceived value theory, and S-O-R theoryPLOS ONE

Dear Dr. Chen,

Thank you for submitting your manuscript to PLOS ONE. After careful consideration, we feel that it has merit but does not fully meet PLOS ONE’s publication criteria as it currently stands. Therefore, we invite you to submit a revised version of the manuscript that addresses the points raised during the review process.

We look forward to receiving your revised manuscript.

Kind regards,

Sudarsan Jayasingh, Ph.D

Academic Editor

PLOS ONE

Journal Requirements:

Reviewers' comments:

Reviewer's Responses to Questions

**Comments to the Author**

1. If the authors have adequately addressed your comments raised in a previous round of review and you feel that this manuscript is now acceptable for publication, you may indicate that here to bypass the “Comments to the Author” section, enter your conflict of interest statement in the “Confidential to Editor” section, and submit your "Accept" recommendation.

Reviewer #1: (No Response)

Reviewer #3: (No Response)

2. Is the manuscript technically sound, and do the data support the conclusions?

Reviewer #1: Yes

Reviewer #3: Partly

3. Has the statistical analysis been performed appropriately and rigorously? 

Reviewer #1: Yes

Reviewer #3: Yes

4. Have the authors made all data underlying the findings in their manuscript fully available?

Reviewer #1: Yes

Reviewer #3: Yes

5. Is the manuscript presented in an intelligible fashion and written in standard English?

Reviewer #1: Yes

Reviewer #3: Yes

6. Review Comments to the Author

Reviewer #1: Upon thorough review of the manuscript titled "Explaining subscription intention for video streaming platforms in China: Integrating the UTAUT2 model, perceived value theory, and S-O-R theory," it has been observed that while the paper attempts to build upon an integrated theoretical framework, the linkage and interplay between the UTAUT2 model, perceived value theory, and S-O-R theory require further clarification. The rationale for the theoretical integration lacks sufficient depth to illustrate how these theories collectively contribute to a more comprehensive understanding of subscription intentions within the Chinese market. Additionally, the operational definitions of key constructs, specifically 'perceived value' and 'attractiveness of alternatives,' may not be adequately tailored to the Chinese cultural context, which is vital for ensuring the relevance and applicability of the findings.

Moreover, the sampling methodology employed raises concerns regarding potential biases and the generalizability of the study's conclusions. The combination of self-selection, volunteer, and convenience sampling methods, while pragmatic, may not provide a sample that is representative of the broader viewer population in China. Consequently, this could limit the external validity of the research. Furthermore, the adoption of measurement scales from various sources necessitates careful cultural adaptation and validation to ensure their appropriateness for the targeted demographic. The data analysis could also benefit from a more in-depth approach, including the exploration of potential differences across demographic groups through multi-group analysis, which the study has not fully leveraged. These aspects are critical for enhancing the robustness and integrity of the study, and addressing them would significantly strengthen the paper's contributions to the field.

Reviewer #3: 1. Introduction

It would be good to indicate why factors such as performance expectancy, effort expectancy, social influence and hedonic motivation are selected to determine intention for video streaming platforms? why UTAUT2 is selected and not other version of technology model such as UTAUT, TAM3 etc. Why SOR and perceived value theory is included? Also, why attractiveness of alternatives is included? It should include some discussion on the context that could strengthen or weaken the relationship.

2. Theoretical background

a. Only UTAUT2 is discussed, how about perceived value theory and SOR?

b. For the proposed hypotheses, the relationship is positive or negative? It is not indicated in the H1 to H5.

c. Habit is not tested for mediation? The research model is not able to be detected in the manuscript.

3. Methodology

a. as it is a combination of self-selection, volunteer, and convenience sampling methods are used for the selection of respondents, it would be good to explain each of the method and how do all these techniques contribute to 506 respondents?

b. How does sample size is determined? indicate the technique.

c. It would be good to include the measures/items for the constructs? also the source of the measures/items.

d. Did any pilot study is performed?

4. Results

In Table 7. Structural model assessment, the direct relationships are presented but is not tested as Hypotheses.

7. PLOS authors have the option to publish the peer review history of their article (what does this mean? ). If published, this will include your full peer review and any attached files.

**Do you want your identity to be public for this peer review?** For information about this choice, including consent withdrawal, please see our Privacy Policy .

Reviewer #1: No

Reviewer #3: No

---

## [Author Response · Author response to Decision Letter 2]

22 Mar 2025

Dear editor and reviewers,

Thanks for your valuable comments. And I revised the manuscript according to your comments.

Editor’s comments :

Authors’ action:

Thanks for your comments. Based on your comments, I rechecked the whole reference list. We deleted the duplicated references and replaced the retracted reference with new reference in the reference list. The reference of “Yoelianto, F., & Utami Tjhin, V. (2022). Social Isolation, A New Variable Affecting Behavioral Intention To Use Subscription Video On Demand. Journal of Theoretical and Applied Information Technology, 100(11),pp.3788-3797. ISSN: 1992-8645.” is retracted reference, which is replaced by the reference (line 8, page .7) of “Tsai, L. L. (2023). A deeper understanding of switching intention and the perceptions of non-subscribers. Information Technology & People, 36(2), 785-807.” In addition, the reference of “22. Endata.(2022). www.endata.com.cn.Endata - Data Intelligence Service Provider_Annual Box Office. [online].2022.https://www.endata.com.cn/BoxOffice/BO/Year/index.html.” is retracted reference, which is replaced by the reference (line 14, page .7) of “22. Huang, Y., Lv, Z., & Sui, Z. (2021, October). Where Should Existing Video Streaming Platforms Improve: A Comparative Analysis of Netflix and IQiyi. In 2021 International Conference on Public Relations and Social Sciences (ICPRSS 2021) (pp. 585-592). Atlantis Press.” Furthermore, the reference of “Ryu, K., & Jang, S. (2008). RETRACTED ARTICLE: Influence of restaurants' physical environments on emotion and behavioral intention. The Service Industries Journal, 28(8), 1151-1165.” is retracted reference, which is replaced by the reference (line 1, page.8) of “Zhang, H., Zheng, S., & Zhu, P. (2024). Why are Indonesian consumers buying on live streaming platforms? Research on consumer perceived value theory. Heliyon, 10(13).”

Review 1 Comments:

1.Upon thorough review of the manuscript titled "Explaining subscription intention for video streaming platforms in China: Integrating the UTAUT2 model, perceived value theory, and S-O-R theory," it has been observed that while the paper attempts to build upon an integrated theoretical framework, the linkage and interplay between the UTAUT2 model, perceived value theory, and S-O-R theory require further clarification. The rationale for the theoretical integration lacks sufficient depth to illustrate how these theories collectively contribute to a more comprehensive understanding of subscription intentions within the Chinese market.

Authors’ action:

Thanks for your comments. Based on your comments, we added contents to further clarification of the linkage and interplay between the UTAUT2 model, perceived value theory, and S-O-R theory in page 4, line 21-29, section of 1. introduction, which indicated as “In UTAUT2 model highlights the factors of performance expectancy, effort expectancy, social influence, hedonic motivation, and habit in shaping users' subscription intentions. In addition, the multi-dimensional aspects of perceived value are used in this research: emotional value, quality value, monetary value, and convenience value by perceivd value theory. Furthermore, based on S-O-R model, this study explores how platform features and factors such as performance expectancy, effort expectancy, social influence and hedonic motivation (i.e., Stimulus) influence users' perceived value (i.e., Organism), subsequently impacting their subscription intentions (i.e., Response) on video streaming platforms. Thus, this research integrates UTAUT2, perceived value theory, and S-O-R model to explain the mechanisms informing subscription intention.” In addition, based on your comments, we added contents to further illustrate how these theories collectively contribute to a more comprehensive understanding of subscription intentions within the Chinese market in page 4, line 40-43; page 5, line 1-5, section of 1. Introduction, which indicated as “Furthermore, new technologies, such as SVOD and artificial intelligence, have changed consumer behavior toward subscription packages in China. SVOD is related to the performance expectancy, effort expectancy, social influence, hedonic motivation, and habit that make consumers to increase their consumption of these platforms as access is unlimited. In addition, consumers in China use these platforms based on their preferences and choices that will help explain their goals. New customers enter these platforms on the basis that they feel attracted to the video content and perceive value during the trial period of the video streaming platform. Thus, this article integrates the UTAUT2 model, perceived value theory, and S-O-R theory to further clarify and contribute to reasons for the unstable subscription numbers of viewers in China. ”

2.Additionally, the operational definitions of key constructs, specifically 'perceived value' and 'attractiveness of alternatives,' may not be adequately tailored to the Chinese cultural context, which is vital for ensuring the relevance and applicability of the findings.

Authors’ action:

Thanks for your comments. Based on your comments, we added the operational definitions of key construct of “perceived value” to adequately tailored to the Chinese cultural context, in page 7, line 25-30, section of 2.2 Mediating role of perceived value, 2. Theoretical background, which indicated as “And regarding to video streaming platforms in China, viewers’ perceived value is very important, which refers to the viewers' comprehensive evaluation of the services, content, experience and other aspects provided by the video streaming platform [49]. For example, in Tencent Video, Iqiyi, Youku and other mainstream video streaming platforms in China, they have continued to enhance the viewers’ perceived value by providing high-quality content, optimising the user experience, strengthening their branding, formulating a reasonable pricing strategy, and improving the quality of their services [50].”

And based on your comments, we added the operational definitions of key construct of “attractiveness of alternatives” to adequately tailored to the Chinese cultural context, in page 8, line 38; page 9, line 1-6, section of 2.3 Moderating role of attractiveness of alternatives, 2. Theoretical background, which indicated as “Video streaming market in China has formed a diversified competitive landscape, including the co-existence of short-form, long-form and live streaming [61]. In addition, the competition among video streaming platforms in China is extremely fierce, and platforms need to continuously innovate and optimize in terms of content quality, user experience, technological innovation and business models to meet the challenges of market competition. Thus, attractiveness of alternatives in video streaming platforms in China plays an important role in viewers’ subscription decisions [62].”

3. The sampling methodology employed raises concerns regarding potential biases and the generalizability of the study's conclusions. The combination of self-selection, volunteer, and convenience sampling methods, while pragmatic, may not provide a sample that is representative of the broader viewer population in China. Consequently, this could limit the external validity of the research.

Authors’ actions:

Thanks for your valuable comments. In this study, volunteer sampling refer to self-selection sampling techniques. Self selection is one of volunteer sampling technique, which is fully aware of all these questions, so we combine volunteer and self-selection to volunteer (self-selection). Because it is online questionnaire, convenience sampling method is suitable for access the online link of questionnaire. This study chooses these to sampling technique has been covered research question. We added the contents in section of page 10, line 15-21, which is indicated as “First of all, the sample size is large enough to cover the research question. Secondly, in the questionnaire, it has screening questions to filter out the issues of non-compliance. In addition, the age group is above 18, which can be covered. Furthermore, for viewers’ demographic, region and employment status, these two sampling technique has been assessed. Thus, in this study combination of volunteer (self-selection) and convenience sampling methods can be applied to this research. Furthermore, previous scholars are also adopt the similar sampling techniques in their studies [8, 15].” For more detailed revision, we revised the contents and added explanation of sampling methods in page 10, line 22-40, section of 3. Methodology, which indicated as “Volunteer sampling is the self-selection of sample members to participate in a study. Furthermore, self-selection sampling is when the study allows each case (usually an individual) to determine their willingness to participate in the study [70]. Self-selection cases are often selected because they have strong feelings or opinions about the research questions or stated goals. Therefore, the study publicized the need for cases through appropriate media advertising or by inviting volunteers to participate. Thus, it chose this sampling technique because some filtering questions (stated in the appendices of questionnaire) were applied to select the most appropriate samples from the target population in order to meet the researcher’s interests regarding the objectives of this study. Additionally, convenience sampling can be used when studying issues such as psychological mechanisms and processes where individual differences are small, the total population is unknown, or random sampling is difficult. This study investigates the influencing factors of subscription intention for VSPs in China, especially about the exploration of the psychological dimension of consumers. Furthermore, for larger survey samples, it is not easy to obtain a complete sample frame. Thus, this study chose convenience sampling to send the questionnaire link as a post through three major social media platforms to facilitate respondents to answer and submit the survey. For sample size, in this research, it uses G*Power (a power probability = 0.95, effect size = 0.15 (medium), alpha error probability = 0.05, number of predictors is 8, the minimum sample size for this study was determined to be 160 respondents [71]. Furthermore, according to Comrey and Lee (2013) conducted a study to assess the sufficiency of sample size for factor analysis, which concludes that an excellent factor analysis program would require a sample size of 1,000 while a sample size of 500 would be considered good [72].”

4.The adoption of measurement scales from various sources necessitates careful cultural adaptation and validation to ensure their appropriateness for the targeted demographic.

Authors’ actions:

Thanks for your valuable comments.

In this study, the measurement scale is adopted from previous studies. But, it is is reliable and validity through construct validity procedure. We added the contents in line 13-19, page 12, section of Methodology, which is indicated as “Additionally, the measurement scale is adopted from previous studies, which is considered valid indicated in Supporting Information_Appendix 1 Questionnaire [11, 21, 31, 46, 51, 60, 73, 74]. In this study, we did construct validity through 2 marketing experts from video stream industries, 3 university scholars, and 6 potential respondents (randomly selected) to ensure the clarity and comprehensibility of the questionnaire prior to pilot test. Then, this study proceed to pilot study. In addition, digital media consumption and subscription has become a global (e.g. Netflix, HBO Max, IQIYI) tread that is more familiar with most consumers.”

5.The data analysis could also benefit from a more in-depth approach, including the exploration of potential differences across demographic groups through multi-group analysis, which the study has not fully leveraged. These aspects are critical for enhancing the robustness and integrity of the study, and addressing them would significantly strengthen the paper's contributions to the field.

Authors’ actions:

Thanks for your valuable comments. We run the multi-group analysis with female and male. And I added the section of Multi-group analysis in line 6-16, page 7, section of 4. Results, which is indicated as “In the final part of the research, we evaluated the significant differences between female and male that existed in the effects of influencing factors on subscription intention (S). In the multi-group comparison, gender had a significant difference in the moderation impact of AOA on the relationship between perceived value (PV) and subscription (S). In addition, the differences in path coefficient of moderation (AoA*PV) revealed that female group has relatively greater impact in comparison to male group (path diff = 0.135, p = 0.023). All the other differences in the hypothesized relationships were found insignificant. The results of the multi-group analysis are summarized in table 10.

Reviewer #3 Comments: 

1.Introduction

It would be good to indicate why factors such as performance expectancy, effort expectancy, social influence and hedonic motivation are selected to determine intention for video streaming platforms? why UTAUT2 is selected and not other version of technology model such as UTAUT, TAM3 etc. Why SOR and perceived value theory is included? Also, why attractiveness of alternatives is included? It should include some discussion on the context that could strengthen or weaken the relationship.

Authors’ actions:

Thanks for your valuable comments. Based on your suggestion, we added the reason why factors such as performance expectancy, effort expectancy, social influence, habit and hedonic motivation are selected to determine subscription intention for video streaming platforms, in page 3, line 1-5, section of 1. Introduction, which is indicated as “SVOD is related to the performance expectancy, effort expectancy, social influence, hedonic motivation, and habit that make consumers increase their consumption of these platforms as access is unlimited. According to literature, performance expectancy, effort expectancy, social influence, hedonic motivation, and habit are based on the Extended Unified Theory of Technology Acceptance and Use (UTAUT2) [11]. ” And based on your comments, we added reasons why UTAUT2 is selected and not other version of technology model such as UTAUT, TAM etc in line 5-14, page 3, section of 1. Introduction, which is indicated as “SVOD is related to the performance expectancy, effort expectancy, social influence, hedonic motivation, and habit that make consumers increase their consumption of these platforms as access is unlimited. According to literature, performance expectancy, effort expectancy, social influence, hedonic motivation, and habit are based on the Extended Unified Theory of Technology Acceptance and Use (UTAUT2) [11]. ” In addition, based on your comments, we added reasons why SOR and perceived value theory is included in line 28-43, page 3; line 1-7, page 4, section of 1. Introduction, which is indicated as “Overall, it is important to look into different aspects of how value is perceived. First, the Theory of Consumer Value (TCV) provides a clear delineation of consumption value that helps determine consumers’ decision-making intentions [20]. Understanding this delineation is important for video streaming platform providers as they can benefit from understanding why individuals use video streaming platforms and why they choose to watch certain content. In addition, studying the value of consumption can also provide users’ service providers with more interactive and engaging con

---

## [Editor Report · Decision Letter 2]

30 Mar 2025

Explaining subscription intention for video streaming platforms in China: Integrating the UTAUT2 model, perceived value theory, and S-O-R theory

PONE-D-24-28668R2

Dear Dr. Chen,

We’re pleased to inform you that your manuscript has been judged scientifically suitable for publication and will be formally accepted for publication once it meets all outstanding technical requirements.

Kind regards,

Sudarsan Jayasingh, Ph.D

Academic Editor

PLOS ONE
---

## [Editor Report · Acceptance letter]

PONE-D-24-28668R2

PLOS ONE

Dear Dr. Chen,

I'm pleased to inform you that your manuscript has been deemed suitable for publication in PLOS ONE. Congratulations! Your manuscript is now being handed over to our production team.

Kind regards,

on behalf of

Dr. Sudarsan Jayasingh

Academic Editor

PLOS ONE